# Roles of Plant Growth-Promoting Rhizobacteria (PGPR) in Stimulating Salinity Stress Defense in Plants: A Review

**DOI:** 10.3390/ijms22063154

**Published:** 2021-03-19

**Authors:** Dung Minh Ha-Tran, Trinh Thi My Nguyen, Shih-Hsun Hung, Eugene Huang, Chieh-Chen Huang

**Affiliations:** 1Molecular and Biological Agricultural Sciences Program, Taiwan International Graduate Program, Academia Sinica and National Chung Hsing University, Taipei 11529, Taiwan; hatranminhdung@gmail.com; 2Department of Life Sciences, National Chung Hsing University, Taichung 40227, Taiwan; mytrinhnguyen0410@gmail.com (T.T.M.N.); walter030170@gmail.com (S.-H.H.); 3Graduate Institute of Biotechnology, National Chung Hsing University, Taichung 40227, Taiwan; 4Department of Horticulture, National Chung Hsing University, Taichung 40227, Taiwan; 5College of Agriculture and Natural Resources, National Chung Hsing University, Taichung 40227, Taiwan; eugenehuang@smail.nchu.edu.tw; 6Innovation and Development Center of Sustainable Agriculture, National Chung Hsing University, Taichung 40227, Taiwan

**Keywords:** PGPR, salt stress, salinity, abiotic stress, ACC deaminase, seed priming, IAA

## Abstract

To date, soil salinity becomes a huge obstacle for food production worldwide since salt stress is one of the major factors limiting agricultural productivity. It is estimated that a significant loss of crops (20–50%) would be due to drought and salinity. To embark upon this harsh situation, numerous strategies such as plant breeding, plant genetic engineering, and a large variety of agricultural practices including the applications of plant growth-promoting rhizobacteria (PGPR) and seed biopriming technique have been developed to improve plant defense system against salt stress, resulting in higher crop yields to meet human’s increasing food demand in the future. In the present review, we update and discuss the advantageous roles of beneficial PGPR as green bioinoculants in mitigating the burden of high saline conditions on morphological parameters and on physio-biochemical attributes of plant crops via diverse mechanisms. In addition, the applications of PGPR as a useful tool in seed biopriming technique are also updated and discussed since this approach exhibits promising potentials in improving seed vigor, rapid seed germination, and seedling growth uniformity. Furthermore, the controversial findings regarding the fluctuation of antioxidants and osmolytes in PGPR-treated plants are also pointed out and discussed.

## 1. Introduction

Soil salinization caused by saline irrigation regimes [1], by water scarcity [2], and by the rise in sea level due to global warming [3]. Another potential source causing soil salinity comes from compost fertilizer since the raw materials for composting operations are food waste and municipal organic waste that contain large quantities of NaCl [4]. Salinity not only hampers crop productivity, but also threatens the sustainability of agro-ecosystems worldwide. The osmotic stress caused by high salinity (100–200 mM) is originated from the reduction in solute potential of soil solution. The reduced solute potential, in turn, leads to the decrease in hydraulic conductance and then in water and solute uptake by plants [5]. This conducts the prevalence of drought-like conditions and makes drought and salinity occur simultaneously in various agricultural systems [6]. Salinity stress also imposes nutrients deficiencies by interfering directly with ion transporters in the root plasma membrane (e.g., K^+^-selective ion channels) [7], and by inhibiting root growth [8,9,10]. Due to the rising severity of salinity on global food production, numerous strategies have been offered to cope with the increasing challenging soil conditions. Along with plant breeding [11], plant genetic engineering [12], and genetic transformation [13], agricultural practices have dramatically contributed to the improvement of plant tolerance to salinity stress. The supplement of calcium (5 mM CaCl_2_) ameliorated the reduction in shoot and root of salt-treated strawberry plants [14]. The pivotal physio–mechanical property of silicon (Si) has been widely noticed in most plants, especially its alleviating role in improving photosynthetic activity, enhancing essential nutrient uptake, and mitigating negative influence of abiotic stress [15]. In the study of Hassanvand et al. (2019) [16], the reduction of pigment content and essential oil yield in geranium (*Pelargonium graveolens*) plants caused by elevated EC levels was effectively ameliorated by a weekly K_2_SiO_3_ application. Green leaf volatiles (GLVs) are an important group of volatile organic compounds (VOCs) emitted by plants under stressful conditions [17], and enable plants to activate defense-related genes [18]. Z-3-hexeny-1-yl acetate (Z-3-HAC), a GLV, was used in seed priming to promote a better salt stress tolerance [17]. The Z-3-HAC-primed peanut (*Arachis hypogaea* L.) seedlings exhibited higher antioxidant enzymes (AEs) activities, a higher net photosynthetic rate (Pn), and an increased osmolyte accumulation, while reduced reactive oxygen species (ROS) levels, electrolyte leakage (EL), and lipid peroxidation (LP) as compared to the non-primed plants [17]. Being a metabolic intermediate in higher plants, 5-aminolevulinic acid (ALA) is a common precursor of tetrapyrroles such as chlorophyll (Chl), heme and siroheme, and this small signaling molecule also participates in several physiological processes to counteract salt stress damage [19]. In the study of Wu et al. (2018) [20], an exogenous application of ALA under salinity increased the contents of intermediates and Chl a, Chl b, as well as repaired the damages of photosynthetic apparatus. Melatonin (N-acetyl-5-methoxytryptamine) (Mel), a ubiquitous multifunctional signaling molecule, functions as a stimulator in several physiochemical responses against stresses [21]. The application of exogenous Mel mitigated salt stress by increasing the contents of polyamines (PAs), the ubiquitous cellular components acting as antistress agents, in wheat seedlings [22]. Moreover, salt tolerance in Mel-treated rice plants was improved via the upregulation of K^+^ transporter genes, the modulation of K^+^ homeostasis and the scavenging of hydroxyl radicals [23].

The bacterization of plant crops with PGPR and the implementation of these useful rhizobacteria in seed biopriming have demonstrated their beneficial properties in enhancing plant growth and development, and in augmenting plant salt stress tolerance through different mechanisms. PGPR aid to alleviate salinity stress in plants by boosting water absorption capability, enhancing essential nutrients uptake, accumulating osmolytes (OS) (e.g., proline (Pro), glutamate (Glu), glycine betaine, soluble sugars, choline, O-sulphate, and polyols), increasing AEs activities (e.g., superoxide dismutase (SOD, EC 1.15.1.1), peroxidase (POD, EC 1.11.1.7), catalase (CAT, EC 1.11.1.6), ascorbate peroxidase (APX, EC 1.11.1.11), monodehydroascorbate reductase (MDAR, EC 1.6.5.4), dehydroascorbate reductase (DHAR, EC 1.8.5.1), glutathione reductase (GR, EC 1.6.4.2), and non-enzymatic antioxidants (NEAs) (e.g., ascorbate (ASC), glutathione (GSH), tocopherols (TCP), carotenoids (Car), and polyphenols (PPs)) in plant tissues [24,25,26,27,28]. In all types of salinity, sodium chloride (NaCl) is the most soluble and widespread salt [17] and Na^+^ is the primary cause of ion-specific damage for many plants, especially for graminaceous crops [29]. Consequently, to narrow down the scope of this review, we focus mainly on the negative effects of Na^+^ ion on plants, although high concentrations of Cl^−^ anion are also toxic to plants. In this point of view, three terms “salt”, “saline”, and “Na^+^” were used interchangeably in the review to indicate the salinity.

## 2. Adverse Effects of Salinity on Plants

### 2.1. Na^+^ Accumulation, Nutrients Uptake Inhibition, and Plant Growth Reduction

Under salt stress, Na^+^ is accumulated at higher concentrations in plant tissues, causing changes in Na^+^/K^+^ ratio and the inhibition of essential nutrient uptake [14,24,30]. This could be attributed to the competition between similar ionic radii of Na^+^ and K^+^ in soils [31], causing the dysfunctional ionic selectivity of the cell membranes. In the review of Manishankar et al. [32], high Na^+^ concentration in soil can change soil texture, leading to a decrease in soil porosity. This leads to the reduction of soil aeration and water conductance. Also according to Manishankar et al. [32], the zones of low water potential caused by high salt deposition in the soil make difficult for the roots to uptake water and nutrients. With 35 mM NaCl treatment, the Na^+^ concentration in strawberry leaves and roots was 3.4-fold higher than that in the control plants [14]. Moreover, salt stress also caused the critical reduction in the fruit yield (FY) with 35% yield loss in the variety Camarosa and 45% in the variety Oso Grande [14]. At 150 mM NaCl, a tremendous increase (50.4-fold) in the Na^+^ content, and an increase in the Na^+^/K^+^ ratio (1.48 vs. 0.02) in the roots of *Broussonetia papyrifera*, a woody plant used in paper industry, was recorded, in harmony with the decrease in K^+^ (25.6%), Ca^2+^ (23.3%), Mg^2+^ (21.4%), and P^3+^ (8.4%) contents [33]. In contrast, an upsurge of Na^+^ concentration was found in the leaves of canola plants (Brassica napus L.), with approximately 4-fold greater than that in the roots [30]. The Na^+^ content in the common bean leaves (*Phaseolus vulgaris* L.) was 5–7–fold higher than that in the control common bean leaves, whereas the K^+^ content was decreased by 32–35% relative to the control plants [34]. Likewise, the Na^+^ content in the salt treated-chickpea leaves (*Cicer arietinum* L.) was 3.2-fold higher than that in the control leaves, leading changes in Na^+^/K^+^ ratio from 0.31 in the non-saline condition to 2.24 in the saline condition. The reduction in N, K, Ca, Mg contents was also recorded by 54%, 55%, 60%, and 55%, respectively as compared to those in the control leaves [35].

In general, phytotoxicity caused by high salt concentrations was found under in vitro and greenhouse conditions and the toxic symptoms increase correlatively with the increase in NaCl treatments. High salinity significantly affects plant growth and physio-biochemical aspects, resulting in the decrease in germination rate (GRA), fresh and dry matters, photosynthetic pigments, essential nutrients uptake, and most importantly, in the loss of final crop yields. In contrast, a significant increase in AEs activities, osmoregulators, LP, membrane damage, ROS contents, Na^+^ accumulation, and Na^+^/K^+^ ratio was obviously observed with the increasing NaCl concentrations [36]. The shoot dry weight (SDW) and root dry weight (RDW) of 35 mM NaCl-treated strawberry plants (*Fragaria* x *ananassa* Duch) were 45.8% and 58.6% lower than those in the control plants, respectively [14]. Salt stress adversely hampers all stages of plant growth, causing the reduction in FY (227 vs. 415 g/plant), fruit weight (FW) (8.4 vs. 9.6 g/fruit), number of fruits per plant (NF) (27 vs. 43), and water-soluble dry matter (SDM) (6.6% vs. 8.4%) of stressed plants relative to the unstressed plants. The root dry weight (RDW) of common bean decreased by 59–61% and the final yield lost by 27–30% under 200 mM NaCl [34]. Similarly, at 200 mM NaCl concentration, salt stress reduced 38% SDW and 50% RDW of chickpea (*Cicer arietinum* cv. Giza 1) compared to the control plants [35]. Regarding the influence of salinity on nutritional values, although moderate saline stress enhanced glucosinolates and antioxidants contents in broccoli (*Brassica oleracea* L. var. *italica* cv. Marathon) (40 mM NaCl) [22], and the application of 6 dS m^−1^ (66 mM NaCl) increased the contents of lycopene, β-carotene, vitamin C and overall phenolic compounds (PCs) of tomato fruits [23], high salinity concentrations (15 dS m^−1^ ~ 200 mM) markedly reduced protein, fat, and crude fiber contents of wheat grains [24]. Moreover, the fruit size of tomato [37] and the FW of pepper [38], which are considered major determinants of price and marketable characteristics, were strongly reduced with the increase of saline levels. However, it is noteworthy that although high salt concentration affects plants in an adverse manner, the definition of “low”, “moderate”, or “high” salinity depends fundamentally on plant variety, growth stage, nutrient composition in soil, and irrigation regime, etc.

### 2.2. Impairment of Physio-Biochemical Attributes

#### 2.2.1. Reduction in Photosynthetic Pigments

Salinity stress causes an unrepairable damage to the photosynthetic apparatus at any development stage of plant’s life as it alters the chloroplasts structure, degrades chloroplast envelope, and triggers chloroplast protrusions [39]. Numerous studies indicated that high salinity led to a serious degradation of Chl and Car in salt-stressed plants; however, the degrees of reduction in these photosynthetic pigments (PhoPs) depended largely on plant species, plant age, NaCl concentration and the duration of salt stress exposure. Specifically, only 9%, 11%, 13%, and 14% reduction in the total chlorophyll (Tchl) were determined in the rice (*Oryza sativa* L.) [40], soybean (*Glycine max* (L.) Merr.) [41], maize (*Zea mays* L.) [42], and cucumber (*Cucumis sativus* L.) [43] seedlings, respectively. However, contrary to these studies, the reduced contents of Tchl were tremendously varied from 22% in oat (*Avena sativa*) seedlings [44], 41–42% in tomato (*Solanum lycopersicum* L.) seedlings [45,46], 44% in tomato (*Solanum lycopersicum* L.) plants [47], 50% in peanut (*Arachis hypogaea*) seedlings [48], 56% in rice (*Oryza sativa* L.) seedlings [49] to 61% in rapeseed (*Brassica napus* L.) plants [30], respectively. In addition, under salt detriment, 16% of Car decreased in the ginseng plantlets, 19% of Car reduced in the mung bean plants, and 49% of Car decreased in the tomato seedings were reported by Sukweenadhi et al. (2018) [50], Shahid et al. (2021) [36], and Akram et al. (2019) [45], respectively. In addition, NaCl toxicity also declined Pn, stomatal conductance, and transpiration rate in the stressed plants [51].

#### 2.2.2. Increase in LP

Lipids are essential components of cell membranes responsible for structure maintenance and cell functions control [52]. ROS are generated from several life processes and an excess of ROS can damage cell, tissues and organs [53]. Salinity exposure brings about a disturbance, an overflow, or even a disruption of electron transport chains (ETC) in mitochondria and chloroplasts in higher plants, resulting in ROS accumulation [54]. The major site involved in the production of O_2_^•−^ is the photosystem I (PSI). In the presence of light, O_2_ which is continuously provided by the water autolysis (Reaction 1: 2H_2_O → 4 e^−^ + O_2_ + 4 H^+^) can be reduced to O_2_^•−^ (Reaction 2: 2O_2_ + 2 e^−^ → 2 O_2_^•−^). The excess amount of reduced ferredoxin (Fd_red_) and the limited NADP availability induce the autoxidation of Fd_red_ to Fd_ox_ and the generation of O_2_^•−^ (Reaction 3: Fd_red_ + O_2_ → Fd_ox_ + O_2_^•−^). In addition, the Fd_red_ can react with O_2_^•−^ to form H_2_O_2_ (Reaction 4: Fd_red_ + O_2_^•−^ + 2 H^+^ → Fd_ox_ + H_2_O_2_). Lipids are primary targets of ROS attack and the free radicals oxidation of polyunsaturated fatty acids is called LP [55]. As a byproduct of LP, malonaldehyde (MDA) has been largely used as an important indicator to evaluate the extent of damaging effects caused by ROS and oxidative stress combination on membrane lipids to reduce membrane stability [56]. The MDA content was tremendously increased by 36% in ginseng root plantlets [50], 39% in maize [42], 47% in peanut [48], 70% in chickpea [35], 131% in oat [44], 153% in mung bean [36], and 300% in rice seedlings [57], indicating an severe damage to cell membrane integrity and/or membrane permeability during salinity exposure [58].

### 2.3. Increased Accumulation of ROS and Elevated Production of AEs, NEAs, and OS

On the one hand, ROS function as signaling molecules to mediate a wide range of important biological processes during plant growth and development such as seed germination [59], cell differentiation [60], root primary growth [61], and stem cell activities [62]. On the other hand, an elevated accumulation of ROS in plant tissues also causes oxidative damage to protein, DNA, lipids, and Chl biosynthesis [63,64]. Salinity stress brings about excessive accumulations of ROS including superoxide radical (O_2_^•−^), hydrogen peroxide (H_2_O_2_), singlet oxygen (^1^O_2_), and hydroxyl radicals (^•^OH), which disturb cellular redox homeostasis and lead to oxidative stress [65]. ROS homeostasis, therefore, is essential to maintain a delicate balance for plant growth, especially under environmentally adverse conditions. To deal with salinity-derived oxidative stress, plants possess enzymatic defense system that synthesizes an array of AEs, along with NEAs to neutralize and detoxify ROS [26,27]. The AEs conduct the scavenging activity by breaking down and removing free radicals, while the NEAs perform their scavenging functions by interrupting free radical chain reactions [66]. Furthermore, the accelerated synthesis and accumulation of OS are also the common responses executed by plants to provide osmotic adjustments and to protect cell membrane integrity [67]. In plants, Pro is synthesized by either glutamate pathway or orinithine pathway [68] and is accumulated in cytosol and vacuole under stressful conditions. Under non-stress conditions, Pro only accounts for less than 5% of the total pool of free amino acids in plants. However, under various stresses, the Pro concentration might increase up to 80% of the total amino acid pool, indicating its vital roles in ROS homeostasis and water balance in plants [69]. Pro was found to exhibit protective roles against damages caused by ^1^O_2_ or ^•^OH [70]. Ethylene (C_2_H_4_), a small volatile phytohormone in higher plants, is involved in all stages of plant growth and development, from seed germination to fruit ripening [71]. Furthermore, ethylene has been considered as a stress hormone since it participates in plant responses to various types of stress such as wounding [72], salinity [73], and drought [74]. Although a small amount of ethylene, which is immediately produced after the onset of a stress, can initiate the systemic resistance in plants, the excess amount of ethylene from the second peak could bring about the inhibition of plant growth or even lead to cell death [75].

## 3. Plant Growth-Promoting Rhizobacteria as the Promising Bioinoculants for Plant Crops

### 3.1. Key Criteria for Being Applicable PGPR

The close alliance among soil, plant, and microbes exists during the entire life cycle of plants promotes plant development, induces systemic resistance in the host plant against pathogens and mitigates salinity stress [76]. PGPR have been widely used for decades to control insects pests [77], plant diseases [78], to promote plant growth [79], to manage nutrient [80], and to alleviate abiotic stress [81]. The ameliorative functions of PGPR consist of three aspects, namely, the ability to protect themselves against hyperosmotic conditions and abnormal NaCl concentrations, the capacity to aid plant tolerate better to elevated salinity, and to improve soil quality [82]. Regarding the alleviating roles of PGPR in promoting plant salinity tolerance, PGPR exhibit beneficial traits in mitigating the toxic effects of high salt concentrations on morphological, physiological, and biochemical processes in plants, resulting in the significant rescue of yield loss. According to Fouda et al. [83], the application of PGPR could ameliorate the negative impacts of salinity via two main mechanisms as follows: (1) PGPR activate stress response systems in the host plants soon after the exposure of the plants to salinity, and (2) PGPR synthesize anti-stress biochemicals such as AEs, NEAs, and OS that are responsible for the removal of ROS [84]. Furthermore, PGPR can also mitigate salt stress symptoms by producing Na^+^-binding exopolysaccharides (EPS), improving ion homeostasis, decreasing ethylene levels through enzyme 1-aminocyclopropane-1-carboxylate (ACC) deaminase, and synthesizing phytohormones [85,86,87].

#### 3.1.1. ACC Deaminase-Producing PGPR and Other Plant Growth Promoting Attributes

Enzyme ACC deaminase [EC 4.1.99.4] catalyzes the cleavage of 1-aminocyclopropane-1-carboxylate (ACC), an intermediate precursor of ethylene in higher plants, to produce α-ketobutyrate and ammonia [88]. A proper amount of ethylene derived from the existing pool of ACC, or so called the small peak of ethylene in the biphasic ethylene response model described by Glick et al. [89] and Pierik et al. [90], is thought to be useful to plants in activating plant defensive responses to stress stimuli (e.g., temperature extremes, drought or flooding, insect pest damages, phytopathogens, and mechanical wounding) [91]. However, an elevated ethylene accumulation, also called stress ethylene or the larger peak of ethylene in the biphasic model, may cause harmful effects (e.g., chlorosis, abscission, and senescence) on plant growth [92], even lead to dead when present at high concentrations in plant tissues [93]. Although PGPR possess many different mechanisms to maintain plant growth under salinity detriment, the production of ACC deaminase is extremely important in reducing the elevated levels of ethylene, thereby indirectly support plant growth. The ACC deaminase-producing PGPR that live on plant surfaces or colonize in the plant tissues function as a sink for ACC [30] and the use of ACC as a nitrogen (N) source is beneficial to plant health since N uptake is always suppressed under salt conditions [94]. Up to now, a plethora of PGPR that have been studied to evaluate their roles in mitigating salinity stress in plants. The PGPR, namely, *Pseudomonas putida* UW4 [30], *Arthrobacter protophormiae* [95], *Enterobacter* sp. EJ01 [96], *Enterobacter* sp. UPMR18 [97], *Zhihengliuella halotolerans*, *Bacillus gibsonii*, *Halomonas* sp. [98], *Chryseobacterium gleum* sp. SUK [99], *Pseudomonas fluorescens* 002 [100], *Microbacterium oleivorans* KNUC7074, *Brevibacterium iodinum* KNUC7183, and *Rhizobium massiliae* KNUC7586 [101], *Stenotrophomonas maltophilia* SBP-9 [102], *Enterobacter* sp. P23 [49], *Burkholderia* sp. MTCC 12259 [57], *Paenibacillus yonginensis* DCY84 [50], *Bacillus pumilus* strain FAB10 [51], *Pantoea agglomerans* [103], *Aneurinibacillus aneurinilyticus* and *Paenibacillus* sp. [88], *Leclercia adecarboxylata* MO1 [104], *Pseudomonas argentinensis* and *Pseudomonas azotoformans* [105], *Bacillus subtilis* (NBRI 28B), *B. subtilis* (NBRI 33 N), *Bacillus safensis* (NBRI 12 M) [106], *Bacillus megaterium* NRCB001, *B. subtilis* subsp. *subtilis* NRCB002, *B. subtilis* NRCB003 [107], and *Kosakonia sacchari* [36] can produce ACC deaminase, as well as other important products such as indole-3-acetic acid (IAA), siderophore (Sid), EPS, and Pro. In addition, PGPR can conduct biofilm forming, N fixation, phosphate (P) solubilization, hydrogen cyanide (HCN) and antifungal enzymes production [99]. The capability of PGPR for moderating salinity damage could be considered an indispensable trait for strain selection [108], reflecting in the elevated amounts of ACC deaminase, IAA, EPS, GSH, and Pro produced by themselves during salt exposure to protect their cells against the damaging effects of high NaCl concentrations. For instance, at 500 mM NaCl, *Sphingomonas* sp. LK11 produced more GSH and Pro to counteract the detrimental effects of salinity imposed on its growth [108]. Similarly, the productions of ACC deaminase and Pro by the halotolerant *Burkholderia* sp. MTCC 12259 were highly correlated with the increasing NaCl concentrations in the medium broth, in which ACC deaminase reached the highest at 600 mM NaCl, while the highest Pro level was obtained at 1000 mM NaCl [57]. This result was in accordance with the report of Ilyas et al. [109] when the Pro produced by a consortium consisting of *Bacillus* sp. (KF719179), *Azospirillum brasilense* (KJ194586), *Azospirillum lipoferum* (KJ434039), and *Pseudomonas stutzeri* (KJ685889) reached the maximum value at the highest NaCl concentration (10%, *w*/*v*). Also, the productions of ROS-quenching enzymes SOD, CAT, POD, PPO, and Pro in *Enterobacter* sp. P23 were increased with the increase in NaCl concentrations [49]. The levels of IAA, Sid, and ACC deaminase produced by *K. sacchari* strain MSK1 were increased with the increasing NaCl concentrations and reached the highest levels at the highest NaCl concentration (400 mM) [36]. Recently, Misra and Chauhan [106] found that two *B. subtilis* strains NBRI 28B, NBRI 33N, and *B. safensis* NBRI 12 M increased the production of ACC deaminase, biofilm, EPS, and Alginate (Alg) in proportion to the increasing NaCl concentrations in nutrient broth. This finding was in corroboration with the previous study of Mukherjee et al. [110], in which *Halomonas* sp. Exo1 could tolerate up to 20% (*w*/*v*) salt concentration and its EPS yield was directly proportional to the increasing NaCl. These findings indicate that to be selected as potential bioinoculants for improving crop yield in saline soil, the PGPR candidates need to possess an ability to withstand and respond appropriately to high salinity in the environment.

#### 3.1.2. Improvements of Growth Parameters, Nutrients Uptake, and Photosynthetic Pigments in PGPR-Inoculated Plants under Non-Stress Conditions

The halotolerant bacterium *Enterobacter* sp. strain P23 isolated from India’s rice fields possesses the abilities to exhibit high ACC deaminase activity, to solubilize P, to produce IAA, Sid, and HCN [49]. In non-tress conditions, the P23-inoculated rice seedlings showed better morphological parameters, namely shoot length (SL), root length (RL), shoot fresh weight (SFW), SDW, root fresh weight (RFW), and RDW, higher Chl content than those in the non-inoculated rice seedlings. This result was consistent with numerous other studies where the PGPR-inoculated plants grew better than the non-inoculated plants in normal environments. Specifically, the values of SFW, RFW, SDW, RDW, Chl a, Chl b, Car, and N, P, and K concentrations in the S20-inoculated maize seedlings were increased by 2%, 6%, 5%, 2%, 4%, 7%, 2%, 16%, 43%, and 2%, respectively, as compared to the control seedlings [111]. Also, in the *Chryseobacterium gleum* sp. SUK + feather lysate inoculum (FLI)-inoculated wheat seedlings, an increase in 24% Tchl, and in 13% amino acids was noticed [99]. Likewise, an increase in SL, RL, SFW, RFW, and Tchl was observed in the *L. adecarboxylata*-inoculated tomato plants with 22%, 16%, 28%, 51%, and 13% higher than those in the control plants, respectively [104]. The same trend in increased vegetative parameters was found in the studies of Li and Jiang [42], Khan et al. [40], Sapre et al. [44], Sarkar et al. [49], Akram et al. [45], and Alexander et al. [48]. The increase in Tchl was widely observed in various studies, however, the extent to which these pigments increased depends on PGPR strains, NaCl treatments, and plant species. For instance, only a 5% Chl increase in maize seedling bacterized with *B. aquimaris* DY-3 was noticed by Li and Jiang (2017) [42], whereas a 12% increase in *P. putida* H-2-3-inoculated soybean seedlings [41], a 17% increase in *S. maltophilia* BJ01-peanut seedlings [48], a 29% increase in *K. sacchari*-treated mung bean seedlings [36], a 41% increase in *Bacillus megaterium* BMA12-bacterized tomato seedlings [45], 46% in *B. pupilus*-inoculated rice seedlings [40], and 60% in *A. brasilense*-treated white clover plants [58].

PGPR can change root-system architecture by producing phytohormones, especially auxins (Aux) [112], volatile compounds [113,114], and by mediating plant ethylene levels via enzyme ACC deaminase [115]. The inoculation of Arabidopsis plants with *Bacillus megaterium* caused a suppression in primary root growth, while induced lateral root growth development, increased lateral root number, and promoted root hair length [116]. Recently, the research group of Chu et al. (2020) [117] also found that *Pseudomonas* PS01 inhibited the elongation of primary roots and triggered the formation of lateral root and the development of root hair. López-Bucio and colleagues [116] suggested that the inhibition of primary root was caused by a decrease in cell elongation and by a reduced cell proliferation in the root meristem. Vegetative parameters of the endophytes-inoculated sorghum plants (*Sorghum bicolor*) were widely varied with different endophytic PGPR species [118]. Intriguingly, although the amounts of IAA produced by *Pseudomonas plecoglossicida*-R382, *Serratia marcescens*-R381, *Pantoea coffeiphila*-R342, *Bacillus cereus*-R8, *Rhodopseudomonas boonkerdii*-R102, and *Nocardioides aromaticivorans*-R21 were comparable, the RDWs of their respective inoculated sorghum plants were significantly different [118]. This finding suggests that besides the effects of the bacterial IAA on root plant architecture, the interactions between plant and microbe are multifaceted and might play a major role in shaping root system development [119].

The positive influences of PGPR treatment on fruit/grain quality, total yield, and marketable grade yield were also investigated. The FY, fruit marketable yield (FMY), FW, fruit length (FL), fruit diameter (FD), and texture of red fruit in *Bacillus subtilis* BEB-13bs-inoculated tomato plants were improved by 21%, 6%, 29%, 9%, and 5%, respectively in comparison with the control plants [120]. The maximum grain yield was recorded in the wheat plants treated with a triple combination of *Bacillus megaterium*, *Enterobacter* sp. and *Arthrobacter chlorophenolicus* [121], as well as the highest nutrient contents (e.g., N, P, Cu, Zn, Mn, and Fe) were observed in the treated wheat grains.

Nevertheless, in some exception cases, the applications of PGPR under normal conditions did not promote plant growth and yield. The PGPR even exhibited negative effects on the growth of eggplant and tomato plants as reported in the studies of Abd El-Azeem et al. [24] and Vaishnav et al., respectively [47]. Specifically, the SFW, SDW, and yield of eggplant were decreased by 8%, 9%, 12%, respectively after inoculated with *X. autotrophicus* BM13, decreased by 12%, 21%, and 30%, respectively when inoculated with *Bacillus brevis* FK2 [24], as well as the SL of *Sphingobacterium* BHU-AV3-inoculated tomato was reduced by 11% [47]. Similarly, the SL, RL, and total plant fresh weight (TPFW) of *C. gleum*-inoculated wheat plants were decreased by 16%, 36%, and 13%, respectively relative to the control [99]. The data in these previous reports were in accordance with our preliminary data (unpublished data) as the SFW and RFW values of the *Curtobacterium* sp. C1-inoculated Arabidopsis plants were lower than those in the uninoculated plants.

Although the suppressive impacts of PGPR on plant growth and yield are scarcely recorded under non-stress conditions, this should be taken into consideration prior to PGPR bacterization practices in field. Furthermore, the response of plant variety to PGPR is genotype-dependent as shown in the report of Nawaz et al. [122] where the salt tolerant wheat genotype Aas-11 responded positively to *Bacillus pumilus* and *Exiguobacterium aurantiacum*, whereas the salt sensitive wheat genotype Galaxy-13 responded better to *Pseudomonas fluorescence*. In this regard, we should agree that the interactions between host plants and microbes are complicated and not always a win–win situation. In addition, the adaptation of plant species to PGPR might markedly vary from case to case due to genetic variation. More investigations at molecular level are required to deeply elucidate the multi-dimensional impacts of microbes on plants.

#### 3.1.3. Improvements of Growth Parameters, Nutrients Uptake, and Photosynthesis in PGPR-Inoculated Plants under Salinity Conditions

Although PGPR can promote plant growth and improve nutrients uptake, as well as stimulate the synthesis of PhoPs in non-stress environments, their ameliorative roles in plant defense responses are fully expressed till plant crops endure harsh environmental conditions. In the reports of Awad et al. [123] and Abd El-Ghany and Attia [124], they found that the bacterization of maize (*Zea mays* L.) plants and faba bean (*Vicia faba* cv. Giza3) seeds with *Azotobacter chroococcum*, an EPS-producing bacterium, had the decreased Na^+^ and Cl^−^ concentrations and the increased N, P, and K concentrations in their plant tissues. PPs, known as potent antioxidants, can eliminate radical species (e.g., ^1^O_2_, O_2_^•−^, OH^−^, H_2_O_2_), thus preventing the propagation of oxidative chain reactions [125]. In the study of Hichem et al. [126], the amounts of total PPs including phenolic acids, flavonoids, anthocyanins and proanthocyanidins increased accordingly with the increased salinity in young and mature maize leaves and the elevated concentrations of these PCs had an inverse correlation with H_2_O_2_ content and LP level in leaves, indicating the scavenging activity of endogenous PCs against free radicals [127]. The total PPs in the leaves of *Azotobacter chroococcum*-inoculated maize seedlings were always higher than those in the non-inoculated maize seedlings, regardless of salt concentrations [128]. Moreover, the total PPs reached the highest level at the highest NaCl treatment (5.85 g NaCl/kg soil). Abd_Allah et al. [35], who evaluated the effects of endophytic *B. subtilis* (BERA71) on mitigating saline soil stress in chickpea plants (*Cicer arietinum* cv. Giza 1), found that the *B. subtilis* (BERA71)-inoculated chickpea plants yielded higher plant biomass, achieved higher photosynthetic pigments, while reduced ROS levels, and LP compared to the non-inoculated seedlings. The positive correlation between Pro accumulation and salt stress adaptation has been widely recognized. However, the results are still controversial, and more investigations should be conducted to thoroughly explain the underlying mechanisms that regulate AEs and OS production.

Regarding nutrient acquisition, the PGPR helped to decrease Na^+^ accumulation, whereas enhanced the acquisition of N, Ca, Mg, and K contents in the chickpea plants [35]. The increased uptake of Mg^2+^ induced by *Bacillus subtilis* and *Bacillus pumilus* inoculation was associated with the elevated PhoPs contents since Mg^2+^ is the major component of Chl [40,129]. Accordingly, the expression level of *Cab2*, the gene encoding a Chl a/b protein in Arabidopsis plant, was downregulated in Mg-deficient plants before any obvious symptom of chlorophyll deficiency appears [130]. However, Abd_Allah and his colleagues [35] did not investigate the mechanisms that enhanced the uptake of essential nutrients. Therefore, it is unclear whether the increased nutrient acquisition in the *B. subtilis*-inoculated chickpea plants was due to the modulation of root architecture [117,131], the mobilization of P in the soil [132,133], or the N fixation [134,135] induced by *B. subtilis*. Similarly, Khan et al. [40] noticed a limited uptake of Na^+^ in *B. pumilus*-inoculated paddy plants, but the fundamental mechanism that suppressed Na^+^ uptake was not thoroughly investigated yet. In contrast, an extensive accumulation of Na^+^ was observed in the shoots of *Bacillus*-inoculated halophyte *Arthrocnemum macrostachyum* under high NaCl concentration (1030 mM) [136]. Up to now, a plenty of studies recognize the roles of PGPR in increasing K^+^/Na^+^ ratio, in activating K^+^-Na^+^ selectivity, in maintaining PhoPs, in enhancing nutrient uptakes, thereby alleviating salt stress in saline environments [40]. However, more studies are needed to clearly elucidate the mechanisms underlying these phenomena. The key findings in recent PGPR studies were presented in Table 1.

#### 3.1.4. Improvements of Growth Parameters, Nutrients Uptake, and Photosynthesis in PGPR-Primed Seeds and Their Respective Seedlings under Salinity Conditions

Seed is a dramatically important component of agricultural production since it is considered the primary determinant in establishing a fruitful crop. Moreover, seed germination is the first and the most critical stages of the plant’s life cycle [137,138]. The uniformity of seed germination is one of the fundamental criteria that is used to evaluate SV [139]. In the era of climate change, seeds always suffer from the environmental challenges that may cause the reduction in seed GRA, GP, and the dysfunction of seedlings, resulting in a decrease in ultimate crop yields. Germinating seeds and seedlings appear to be more sensible to salinity than the growing plants since the germination stage occurs on saline soil surface where the drought-like condition reduces SV, suppresses protein synthesis, and disturbs structural organization in germinating embryos [140,141]. In addition, seed germination is strongly associated with the seedlings survival rate, as well as the subsequent vegetative growth [142]. α-amylase is a key player in starch hydrolysis during seed germination since it supplies carbon source and energy to germinating seeds in the early stages of development before the initiation of the photosynthetic machinery [137]. A reduced water uptake and a decrease in α-amylase activity caused by NaCl might cause the delay of germination time [143]. Furthermore, the data from Dehnavi et al. (2020) [138] demonstrated that salinity accounted for 98% of the variation in tested parameters including GP, germination index, mean germination time, SVI, SL, and RL of seedlings, fresh and dry weight of seedlings, and salinity tolerance indices.

Seed biopriming with living PGPR inoculum stimulates a speed and an uniformity of gemination, assures a rapid, uniform, and high establishment of crops, thereby improving yield and fruit/grain quality in non-stress and harsh conditions [144]. Under non-stress conditions, the GRAs of two endangered fir plant species *Abies hickelii* and *Abies religiosa* were highly stimulated by a combination of 12 h-hydropriming with PGPR biopriming, resulting an improved GRA up to 91% of *P. fluorescens* JUV8-primed *A. hickelli* seeds vs. 28% of unprimed control and up to 68% of *B. subtilis* BsUV-primed *A. religiosa* seeds vs. 32% of unprimed control [145]. Similarly, the GRA of isolate Ac26-primed wheat seeds was increased to 93.3% and the vigor index was 2830.7, much higher than those of the unprimed control with 53.3% and 1097.5, respectively [146]. The subsequent development of primed plants was also better than the unprimed plants, suggesting the lasting impacts of PGPR treatment on physio–biochemical attributes of the treated plants [145,146].

In the study of Sarkar et al. [57], the inoculation of rice seeds with *Enterobacter* sp. strain P23 promoted higher germination percentage (GP) (76% ± 7.03 vs. 48% ± 4.78), and higher seedling vigor index (SVI) (881.6 ± 67 vs. 57.5 ± 12.6) as compared to the non-inoculated seeds. Under salt conditions, the Pro peaked its highest level, the SOD, CAT, POD, PPO, and MDA exhibited their highest contents in uninoculated rice seedlings. However, the activities of these enzymes in P23-inoculated seedlings were significantly reduced relative to those in the non-inoculated seedlings. The productions of ethylene in non-inoculated seedlings and P23 AcdS mutant-inoculated seedlings were comparable, consistent with the study of Cheng et al. [30], while ethylene in the WT P23 strain-treated plants was lower, indicating that the WT P23 succeeded in decreasing stress ethylene production. Under 250 mM NaCl treatment, the SFW and SDW of *P. putida* UW 4-inoculated canola plants (*Brassica napus* L.) were 1.7-fold higher than those of untreated plants [30]. However, the *P. putida* ACC deaminase (AcdS) minus mutant-inoculated canola plants did not show significant difference in SFW and SDW relative to the untreated plants, indicating the critical role of a functional ACC deaminase enzyme in plant growth under salinity stress. The proteins involved in photosynthesis in the WT *P. putida* UW4 plants were downregulated; however, to a lesser extent as compared to that in the uninoculated plants or in the *P. putida* AcdS plants, resulting in the higher chlorophyll contents relative to the uninoculated plants. Surprisingly, both AcdS and WT *P. putida* plants could accumulate large amount of NaCl in their shoots with 3.7–7-fold higher than that in the uninoculated plants, respectively while being able to maintain their normal growth. This could be partly explained by the increase cell permeability caused by IAA that was produced by the WT *P. putida* and the AcdS mutant. This finding is intriguing and controversial since numerous other studies recognized the decreased Na^+^ uptake in PGPR-bacterized plants [40,42,99,111]. In their another study, Sarkar et al. (2018) [49] primed the rice seeds (*Oryza sativa* cv. Ratna) with *Enterobacter* sp. P23 and achieved greater GRA (76% vs. 48%), as well as SVI (881.6 vs. 57.6) relative to the unprimed seeds. Subsequently, the growth and development of the primed seedlings were better than the unprimed control, representing via greater SFW, RFW, SDW, RDW, SL, RL, amylase, protease, Aux, and Chl values [49,57]. In the study of Zhu et al. (2020) [107], the treatment with 130 mM NaCl severely affected the GRA of the non-primed alfalfa seeds (*Medicago sativa* L.) in comparison with the primed seeds. Specifically, the gemination rate of the non-primed seeds reduced to 29% versus 32% of *B. megaterium* NRCB001-primed seeds, 42% of *B. subtilis* NRCB002, and 40% of *B. subtilis* NRCB003. Also in Zhu et al. [107], the vegetative parameters such as PH, RL, NL, TLA, and TPDW of primed seedlings were always higher than those of unprimed seedlings and the MDA content in their leaves were lower, suggesting a less injured cellular membrane in the primed alfalfa grass.

Regarding the synergy between different PGPR and/or between the microbes and chemicals, the synergistic effects of a consortium (*A. aneurinilyticus* + *Paenibacillus* sp.) were observed via the maximum physio-morphology parameters of primed French bean seedlings (*Phaseolus vulgaris*) in comparison to uninoculated- or individual *A. aneurinilyticus* and *Paenibacillus*-primed seedlings [88]. Two VOCs, namely, 4-nitroguaiacol and quinoline derived from *Pseudomonas simiae* exhibited their ability to induce soybean seed germination under 100 mM NaCl treatment [147]. Furthermore, the combined treatment of sodium nitroprusside (SNP) and *P. simiae* resulted in the higher biomass, the lower MDA content and EL in the treated soybean plants than other treatment plants [147]. Mel exhibits pleiotropic biological activities such as growth regulation [148] and antioxidative property [149] and has been widely used as a promising tool for mitigating salt stress in plants. Abd El-Ghany and Attia (2020) [124] found that the combination of Mel and peat-based inoculants (*Rhizobium leguminosarum*, a N fixing bacterium, and *Azotobacter chroococcum*, an EPS-producing bacterium) synergistically enhanced salt stress tolerance in faba bean plants (*Vicia faba*) as compared to Mel- or inoculants-treated seeds alone. Specifically, in the combined treatment (100 µM Mel + inoculants), the content of Chl a, Chl b, Car, and Pro reached the highest, suggesting the synergistic effects of Mel and beneficial PGPR in improving plant growth and other physiological aspects in salt stress conditions. The combination of Mel and bacterial inoculants, in contrast, helped to boost the faba bean plant growth, to increase PhoPs, Pro, N–P–K uptake, and to reduce the Na^+^/K^+^ ratio.

In conclusion, seed biopriming using PGPR enhances the GRA and SV index in the primed seeds as compared to the unprimed seeds under saline conditions, thereby supporting plants a vigorous growth and a better salinity tolerance during their whole life [150]. The key findings in recent seed biopriming studies were presented in Table 2.

### 3.2. The Increase in AEs and/or Osmoregulators in PGPR-Inoculated Plants and PGPR-Primed Seedlings under Salt Stress

The increased activities of AEs and/or the elevated accumulations of osmoregulators in PGPB-inoculated plants were reported by Li and Jiang [42], Akram et al. [45], Vaishnav et al. [47], Khalid et al. [58], Kim et al. [96], Habib et al. [97], Kang et al. [104], Zhu et al. [107], Halo et al. [151], Bharti et al. [152], El-Esawi et al. [153], Vimal et al. [154], El-Nahrawy and Yassin [155], Sun et al. [156]. For instance, the activity of ROS-scavenging enzymes SOD, CAT of *Enterobacter*-treated okra plants was the highest amongst all treatments, in parallel with their highest vegetative parameters SFW, SDW, RFW, and RDW [97]. Likewise, APX activity in *Enterobacter*-inoculated tomato plants was 20% higher and 2,2-diphenyl-1-picryl-hydrazyl-hydrate (DPPH) assay showed 24% increase in scavenging capacity in the inoculated plants relative to the control plants [96]. In the study of Abd_Allah et al. [35], the activities of POD, CAT, GR, and SOD, and the contents of AsA, GSH and proline were always the highest in the inoculated chickpea plants. The Arabidopsis plants inoculated with *Burkholderia phytofirmans* PsJN revealed an elevated Pro accumulation in comparison with the control plants [157]. In the *Leclercia adcarboxylata*-treated tomato plants, Pro, serine (Ser), glycine (Gly), methionine (Met), and threonine (Thr), as well as citric acid (CA) and malic acid (MA) were significantly accumulated [104]. The detailed profiles of AEs were presented in Table 1 and Table 2.

### 3.3. The Reduction in AEs and OS in PGPR-Inoculated Plants and PGPR-Primed Seedlings under Salt Stress

The changes in AEs and osmo-regulators have been noticed in both uninoculated- and inoculated plants under normal and salinity conditions. However, the reduction or increase of these enzymes in PGPB-inoculated plants in response to salt conditions remains controversial. The decreased profiles of OS and/or ROS-scavenging enzymes were remarked by Kang et al. (2014 a) [43], Kang et al. (2014 b) [41], Barnawal et al. (2014) [95], Khan et al. (2016) [40], Bhise et al. (2017) [99], Abd_Allah et al. (2018) [35], Sapre et al. (2018) [44], Ansari et al. (2019) [51], Alexander et al. (2020) [48], Misra and Chauhan (2020) [106], and Shahid et al. (2021) [36]. Specifically, in the study of Kang et al. (2014a) [43], the activities of CAT, PPO, and POD enzymes and the PPs contents in the inoculated plants (e.g., *B. cepacia* SE4, *Promicromonospora* sp. SE188 or *A. calcoaceticus* SE370) were lower than those in the uninoculated plants under salt stress (120 mM of NaCl). The reduced profiles of AEs in Kang and his colleagues’ findings were in agreement with their another study on soybean using the bacterium *P. putida* H-2-3 [41] and also in line with the study of Sapre et al. [44]. According to Sapre and colleagues’ finding, the *Klebsiella*-treated wheat plants increased by 96% SOD and 286% POD, while the SOD and POD in untreated plants were increased by 353% and 540%, respectively. These data were in agreement with those found by Shahid et al. [36], and Sarkar et al. [49] as these investigators found that the highest antioxidant enzyme activities were recorded in the non-inoculated mung bean and rice plants, respectively. In parallel with the report of Sarkar et al. [49], Rojas-Tapias et al. [128] also recorded the highest Pro content was found in the non-inoculated maize seedling leaves under salt stress. The increase of PP contents in bacterized plants was also recorded in [41,43], however, to a lesser extent than those in the untreated plants. Similarly, Pro accumulations in the tissues of the control oat plants and the control rice plants were much higher than those in the *Klebsiella*-inoculated oat plants and *Entorobacter*-inoculated rice plants (230% and 175%, respectively vs. 155% and 75%, respectively) [44,49]. These studies showed similar findings with Manaf and Zayed [158] as the SOD activity and proline content in the cowpea plants treated with mycorrhizae or *P. fluorescence* alone were lower than those in the untreated plants under 3000 ppm NaCl irrigation regime. Manaf and Zayed assumed that the harmful effects of high salinity made the plants lose the ability to control their metabolites [158], whereas Sapre et al. [44] speculated that the treated plants did not sense much stress as the untreated plants did, leading the lower levels of AEs, NEAs, and osmoregulators in their tissues. Misra and Chauhan [106] proposed that the reduced Pro and AEs in *Bacillus*-treated maize plants may be due to the formation of EPS and biofilm on plant root surfaces that prevented plants from over-uptake Na^+^, thereby attenuating the detrimental effects of toxic ions on plants. This assumption was corroborated by a study of Mukherjee et al. [110], who found that the amount of EPS-bound Na^+^ increased with the increase in NaCl concentration in the solution, thus confirming an efficient role of EPS in NaCl sequestration. In addition, Sarkar et al. [49] explained that the increased antioxidant enzyme activities of *Enterobacter* sp. P23 under saline stress could indirectly quench a significant amount of ROS in rice seedlings, thus delaying the urge to synthesize ROS scavengers by stressed plants.

### 3.4. Genetic Diversities of Plant and Microbe, Plant–Microbe Interactions and Microbe–Microbe Interactions Are Key Players in Regulating AEs Profiles

In the first case where different plant species inoculated with PGPR species from the same genus *Curtobacterium*, the reduced PPO and POD activities were observed in the *Curtobacterium oceanosedimentum* SAK1-treated soybean plants [159], whereas the increase in POD, CAT, SOD, and APX were recorded in the *Curtobacterium albidum* SRV4-treated paddy plants [154].

In the second situation, soybean plants inoculated with different PGPR species also revealed the contrasting antioxidant enzyme profiles. For instance, the soybean plants cv. Giza 35 treated with *Bacillus firmus* SW5 showed a significant increase in APX, SOD, CAT, and POD activities [153], whereas the SOD and DPPH scavenging activities were relatively decreased in the soybean plants cv. Taekwang inoculated with *Pseudomonas putida* H-2-3 [41]. Moreover, variation in enzyme activities were found in the maize variety cv. Maharaja inoculated with different PGPR species [106]. Specifically, the maize seedlings Maharaja treated with *Bacillus subtilis* (NBNI 28B) had higher GPX and CAT as compared to the control, whereas the seedlings treated with *B. subtilis* (NBRI 33 N) and *B. safensis* (NBRI 12 M) exhibited lower SOD, APX, GPX, CAT, and PPO than those in the control and those in the NBNI 28B-bacterized seedlings [106].

Regarding the effect of consortium treatment on AEs, the POD activity in the salt-sensitive wheat genotype Galaxy-13 inoculated with individual *Pseudomonas fluorescence*, *Bacillus pumilus*, and *Exiguobacterium aurantiacum* was always higher than that in the salt-tolerant wheat genotype Aas-11 [122]. The effect of the consortium (*Pseudomonas fluorescence*, *Bacillus pumilus*, and *Exiguobacterium aurantiacum*), however, resulted in the lower POD activity in the treated Galaxy-13 in respect of the treated Aas-11. Similarly, CAT activity was higher in the Galaxy-13 treated alone with *P. fluorescence* and *E. aurantiacum*, in comparison with that in Aas-11, but the CAT activity of Galaxy-13 was lower than that of Aas-11 in the consortium treatment. In contrast, Galaxy-13 had lower SOD activity in a single inoculation with *P. fluorescence* and *B. pumilus*, but it exhibited higher SOD activity than Aas-11 in the consortium treatment.

It is worthy to note that antioxidant response to salinity was varied in different cultivars in the same plant species. Kharusi et al. (2019) [160] noticed that the salt-tolerant date palm cultivar Umsila maintained a normal concentration of ROS by accumulating elevated NEAs and by stimulating greater AEs activities with respect to the salt-sensitive date palm cultivar Zabad. The activities of SOD, CAT, APX and the contents of GSH, FLA, PCs, and Pro in Umsila were statistically significantly greater than those in Zabad when exposed to salt stress.

In summary, the findings in these previous studies, taken together, suggest that an increase or a reduction in the activities of AEs and/or OS in PGPR-inoculated plants during salt stress adaptation depends mainly on the specificities of plant species, on PGPR species, interactions between PGPR in consortia, and on plant–microbe interactions. These controversial data indicate not only that the fine-tuning of the ROS quenchers might be critical for plants to tolerate better to salt stress, but also pose questions concerning the exact mechanisms of salt stress tolerance imposed by PGPR. So far, investigators mainly based on their personal assumption, but not on scientific evidence, to elucidate the fluctuation in AEs. Integrated Omics approach would be necessary to gain insight into this interesting issue. The main message of the present review was displayed in the Figure 1.

## 4. Roles of Multi-Omics Techniques in Deciphering Plant–Microbe Interactions

The modes of action of PGPR on plant salt-stress response mechanism are diverse and complex and remain largely unclear, especially at the molecular level. In the study of Kim et al. (2014) [96], the colonization of *Enterobacter* sp. EJ01 in Arabidopsis root tissues conferred salt stress resistance by inducing salt stress responsive signaling pathways. Specifically, after EJ01 inoculation, the expression levels of *DREB2b*, *RD29A*, and *RAB18* genes related to ABA-dependent and ABA-independent pathways were upregulated, even in the absence of salinity. The expression pattern of *RD29B*, however, was dependent on salt treatment. In addition, *P5CS1*, a Pro biosynthesis-related gene, was upregulated under salinity conditions. The inoculation of EJ01 into Arabidopsis plants induced the host basal innate immunity, as well as the rapid defense responses at systemic level so called induced systemic resistance (ISR). PGPR-elicited ISR was previously observed in Arabidopsis seedlings treated with VOCs from *Bacillus subtilis* GB03 and *Bacillus amyloliquefaciens* IN937a. The use of transgenic and mutant lines of Arabidopsis indicated that the ISR activated by the VOCs from GB03 was based on ethylene-dependent signaling pathway, whereas the ISR was triggered by VOCs from IN937a through an ethylene-independent signaling pathway [161]. In the study of Chen et al. [162], the upregulation of Na^+^/H^+^ antiporter (*NHX*) and *H*^+^*-PPase* genes in *Bacillus amyloliquefaciens* SQR9-inoculated maize shoots facilitated Na^+^ sequestration into vacuoles. The recirculating of Na^+^ from shoot to root via an elevated expression of high-affinity K^+^ transporter 1 (*HKT1*) gene was observed in the treated maize plants. However, contrary to the results of Chen et al. [162], the lowest expression pattern of *HKT1* gene was found in the *Pseudomonas simiae* + sodium nitroprusside (SNP) treated soybean plants [147]. A stable photosynthetic activity was maintained by the highly expressed *RBCS* (RubisCo small subunit), *RBCL* (Rubisco large subunit) genes [162], similar to the upregulation of the Rubisco-encoding gene *rbcL* in the *Klebsiella*-treated oat seedlings [44]. Moreover, the senescence rate in SQR9-inoculated treated maize was properly controlled by the reduced expression of *NCED*, a key gene in ABA synthesis pathway.

In all treatments, the expression levels of AEs-encoding genes *POD* and *CAT* were the highest in the *P. simiae* + SNP-treated soybean plants [147]. Genes associated with Aux, CK, and GA signaling pathways in the *Paenibacillus polymyxa* YC0136-treated tobacco plants (*Nicotiana tabacum* L.) were found to be upregulated relative to the uninoculated control plants, along with the elevated expression of WRKY and MYB transcription factors (TFs) [163]. The WRKY and MYB TFs are responsible for gene regulations and have a great influence in every aspect of plant growth and development, as well as in plant stress responses [164,165]. Changes in the expression pattern of WRKY TF gene under salinity conditions were reported in previous studies [166,167,168]. These findings were in line with the transcriptome profiles of the rice roots bacterized with *Azospirillum brasilense*, in which the hormones-encoding genes (e.g., Aux efflux carriers, Aux-responsive genes, Aux response factors, ACC oxidase genes, ethylene insensitive 2, cytokinin-O-glucosyltransferases, and cytokinin dehydrogenase precursors) were significantly upregulated, as well as the major plant TFs families, namely, AP2/ERF family, MYB family, WRKY family, and the GRAS family [169]. The enhanced expressions of MYB and WRKY TFs were also noticed in the *Dietzia natronolimnaea*-inoculated wheat plants under salinity stress [152]. Furthermore, 9 genes in the SA pathway and 6 genes encoding phenylalanine ammonia lyase (PAL), a key enzyme in the phenylpropanoids metabolic pathway, were upregulated with respect to the control plants, resulting the induction of systemic resistance in tobacco host plant [163]. Malviya et al. [170] also found that the infection of *Burkholderia anthina* MYSP113 into the sugarcane plantlets cv. GXB-9 induced the upregulation of phenylpropanoid biosynthesis genes and amino acid biosynthesis pathways, in accordance with the findings from Liu et al. [163]. Likewise, phenylpropanoid biosynthesis was the most enriched pathway in both differentially expressed genes (DEGs) and abundant metabolites (DAMs) among all salt stress responses in barley rootzones [171]. In the *Arthrobacter nitroguajacolicus*-inoculated wheat roots treated with 200 mM NaCl, 8 genes responsible for Fe uptake, and 2 phosphatase-encoding genes were upregulated, as well as the upregulation of several transporter genes which were in charge of ions, sugars, oligopeptide, and amino acids transports [172].

On the one hand, PGPR influence the expression patterns in the host plants. On the other hand, changes in their transcriptome in response to their colonized plants were also recorded [173]. During the interaction with the host plant, 2 genes *ilvB* and *PPYC1_23850* related to Aux biosynthesis, 3 genes belonging to cell motility category (e.g., *fliG*, *fliH*, *fliF*), 31 genes related to transport proteins including 16 genes belong to ATP-binding cassette (ABC), and 3 genes associated with a major facilitator superfamily (MFS) in *P. polymyxa* YC0136 were significantly upregulated [163]. It was thought that root exudates from tobacco attracted *P. polymyxa* YC0136 and may play roles as nutrients source for the growth of YC0136 strain. This explained the upregulation of numerous transport and cell motility genes in the bacterium. In response to host rice seedlings, *Bacillus subtilis* OKB105 also altered its transcriptomic patterns, in which 52 genes related to nutrients transport and metabolism were upregulated, suggesting the bacterium used carbohydrates and amino acids secreted by rice seedlings as carbon and energy sources. In contrast to the data in [163], many genes involved in chemotaxis and motility, however, were downregulated [173].

In summary, plants and PGPR influence each other in a mutualistic relationship. Regarding plant resistance to salinity, the microbes regulate the WRKY and MYB TFs which are widely distributed in higher plants. Subsequently, these master regulators will regulate the expression of their key downstream stress responsive genes. The plants, in turn, provide nutrients via root exudates for the growth of the microbes. This interaction benefits plants in non-stress conditions, and also in environmental challenging conditions.

## 5. Promise, Limitations, and Future Directions

Considerable PGPB-related studies that have been carried out in the last decades help to improve our knowledge concerning advantageous characteristics of PGPB, in both basic and applied aspects. However, most studies focused on estimating the parameters in vegetative growth stage, but rarely on evaluating the parameters that are related to reproductive stage such as GW and FW, numbers of flower, numbers of seed, fruit per plant, and plant yield. We found a scarcity of studies that evaluated beneficial effects of PGPB on attenuating yield loss and on improving nutrient values. In our opinion, this could be one of the main drawbacks of PGPB-related studies thus far if we consider the improvement of crop yields, productivity, and the quality of fruit/grain under high saline conditions to be our main goal in plant agriculture studies. In addition, in some studies, the lack of important measurements regarding ion contents, ROS levels, phytohormone concentrations, and electrolyte leakage in many studies make them difficult to evaluate the overall effects of PGPB on plants. Furthermore, the short exposure of plants to salinity in some studies unlikely reflects the real situation in fields where a variety of biotic and abiotic stresses endures simultaneously and lasts permanently.

The recognition of PGPR as safe, efficient, and appropriate bioinoculants for agricultural practice is widely accorded. However, the primary mechanisms employed by PGPR to promote plant defense against salt stress need to be deeply unraveled, especially changes in both Omics profiles (e.g., proteomics, transcriptomics, and metabolomics) in the treated plants and in the microbes during interaction with their hosts. In addition, the highly genetic variations of plants and PGPR are useful traits in coping with diverse environmental issues. However, this attribute also makes the reproducibility from previous studies’ findings challenging. As discussed in the present review, the patterns of ROS quencher in PGPR-treated plants exhibited great differences from case to case mainly due to the genetic diversity. The synergistic and/or antagonistic effects between PGPR in consortia on plant growth and defense system, which occur commonly in terrestrial soil ecosystems, should also be thoroughly deciphered. Consequently, an integration of Omics technologies and systems biology should be considered in future studies to provide broader picture and more detailed information concerning plant–microbe interactions in a more complex scenario.

## Figures and Tables

**Figure 1 ijms-22-03154-f001:**
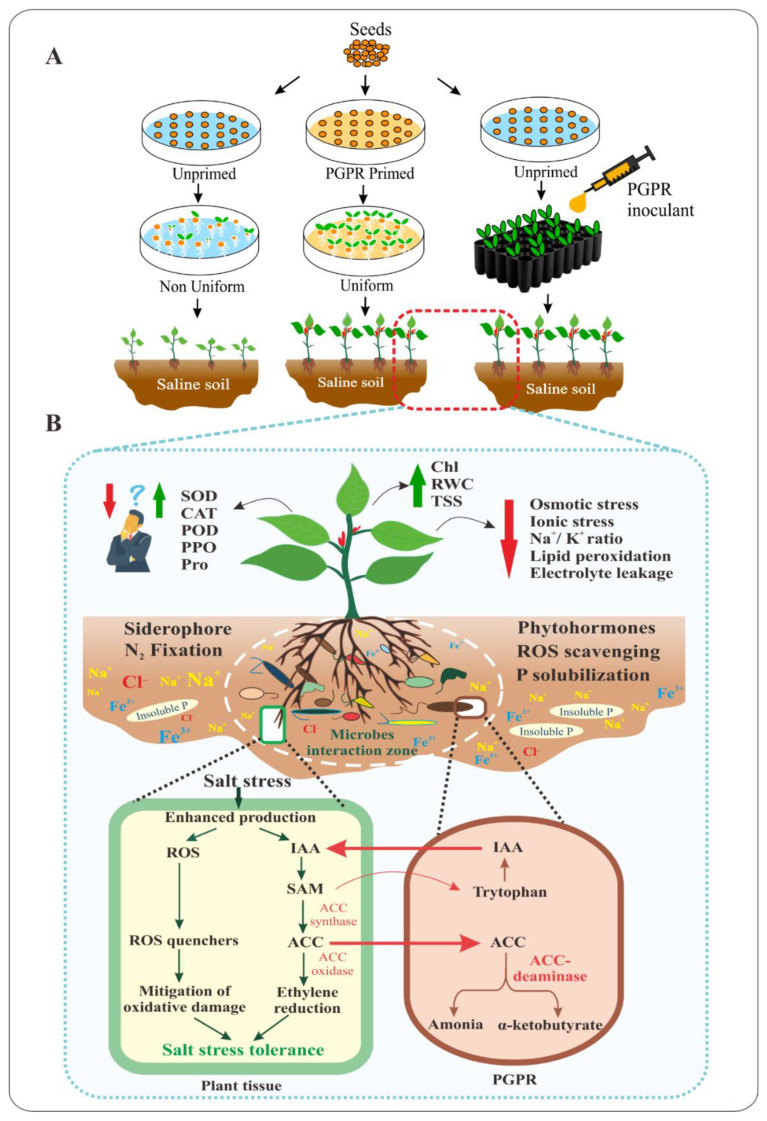
Roles of PGPR in alleviating salinity stress in plants. (**A**) represents the application of PGPR as microbial beneficial tools in seed biopriming technique and as green bioinoculants in seedlings treatment. The primed seeds demonstrate rapid germination and robust, uniform seedlings. (**B**) shows positive effects of PGPR on vegetative parameters and physio-biochemical indexes in PGPR-inoculated plants via various mechanisms e.g., production of OS, AEs to reduce osmotic and ionic stress, and EPS suppress toxic ions uptake and ion exposure. The fluctuation of AEs and OS profiles in PGPR-treated plants is also displayed in the left panel. The middle panel demonstrates key characteristics of PGPR including the production of Sid, phytohormones, EPS, N fixation and P solubilization. The lower panel emphasizes the importance of ACC deaminase-producing PGPR in ameliorating the inhibitory effects of excess ethylene on plant growth.

**Table 1 ijms-22-03154-t001:** Ameliorative effects of plant growth-promoting rhizobacteria (PGPR) on plant growth and physio-biochemical parameters under salinity conditions.

PGPR	Treatments	GP	Hormones	PhoPs	MDA	AEs	NEAs	Pro	Ion Content	Sources
*B. cepacia* SE4, *Promicromonospora* SE188, and *A. calcoaceticus* SE370. 7-day-old tomato seedlings inoculated with PGPR.	Control									[43]
	120 mM NaCl + Uninoculated	↓ 17% SFW ↓ 25% SDW	↑ 255% ABA, ↑ 194% SA	↓ 14% Tchl	N/A	↑ 86% CAT ↑ 213% POD ↑ 456% PPO	↑ 79% PP	N/A	In shoot: ↑ 740% Na^+^ ↓ 4% K^+^ Na^+^/K^+^ ratio ~0.28	
	120 mM NaCl + SE4	↓ 11% SFW ↓ 8% SDW	↑ 10% ABA ↑ 367% SA	↓ 0% Tchl		↑ 27% CAT ↑ 163% POD ↑ 333% PPO	↑ 45% PP		In shoot: ↑ 297% Na^+^ ↑ 17% K^+^ Na^+^/K^+^ ratio ~0.11	
	120 mM NaCl + SE118	↓ 13% SFW ↓ 6% SDW	↑ 6% ABA ↑ 217% SA	↓ 0% Tchl		↑ 23% CAT ↑ 131% POD ↑ 322% PPO	↑ 35% PP			
	120 mM NaCl + SE370	↓ 10% SFW ↓ 9% SDW	↑ 23% ABA, ↑ 261% SA	↓ 0% Tchl		↑ 46% CAT ↑ 156% POD ↑ 322% PPO	↑ 52% PP			
*P. putida* H-2-3. 21-day-old soybean seedlings inoculated with *P. putida.*	Control									[41]
	120 mM NaCl + Uninoculated	↓ 18% SL ↓ 12% TPFW	↑ 33% ABA ↑ 114% SA ↓ 11% JA	↓ 11% Tchl	N/A	↑ 301% SOD	↓ 23% total PP	N/A	In whole plant: ↑ 86% Na^+^, ↑ 55% P	
	0 mM NaCl + *P. putida*	↑ 17% SL ↑ 8% TPFW	↑ 18% ABA ↑ 29% SA ↓ 25% JA	↑ 12% Tchl		↑ 2% SOD	Unchanged total PP		In whole plant: ↓ 17% Na^+^, ↑ 22% P	
	120 mM NaCl + *P. putida*	↓ 9% SL ↓ 0% TPFW	↓ 6% ABA ↓ 26% SA ↑ 54% JA	↓ 7% Tchl		↑ 4% SOD	↑ 4% total PP		In whole plant: ↑ 45% Na^+^, ↑ 30% P	
*B. pumilus.* 14-day-old rice seedlings inoculated with *B. pumilus*.	Control									[40]
	0 mM NaCl + *B. pumilus*	↑ 22% SFW	N/A	↑ 46% Tchl	N/A	↑ 22% SOD ↑ 20% POD ↑ 73% CAT	N/A	↑ 7%	In shoot: ↓ 54% Na^+^, ↑ 57% K^+^, ↑ 76% Mg^2+^, ↑ 18% Ca^2+^, Na^+^/K^+^ ratio ~0.27	
	10 ppm Boron + Uninoculated	↓ 0% SFW		↓ 18% Tchl		↑ 274% SOD ↑ 212% POD ↑ 204% CAT		↑ 41%	In shoot: ↓ 23% Na^+^, ↑ 7% K^+^, ↑ 5% Mg^2+^, ↑ 0% Ca^2+^, Na^+^/K^+^ ratio ~0.67	
	10 ppm Boron + Inoculated	↑ 18% SFW		↑ 59% Tchl		↑ 400% SOD ↑ 272% POD ↑ 254% CAT		↑ 74%	In shoot: ↓ 31% Na^+^, ↑ 61% K^+^, ↑ 67% Mg^2+^, ↑ 18% Ca^2+^, Na^+^/K^+^ ratio ~0.4	
	150 mM NaCl + Uninoculated	↓ 10% SFW		↓ 9% Tchl		↑ 248% SOD ↑ 168% POD ↑ 204% CAT		↑ 56%	In shoot: ↑ 458% Na^+^, ↓ 50% K^+^, ↓ 38% Mg^2+^, ↓ 76% Ca^2+^, Na^+^/K^+^ ratio ~10.4	
	150 mM NaCl + *B. pumilus*	↑ 11% SFW		↑ 86% Tchl		↑ 348% SOD ↑ 220% POD ↑ 273% CAT		↑ 83%	In shoot: ↑ 185% Na^+^, ↑ 24% Mg^2+^, ↓ 7% K^+^, ↓ 18% Ca^2+^, Na^+^/K^+^ ratio ~3	
	10 ppm Boron + 150 mM NaCl + Uninoculated	↓ 10% SFW		↓ 23% Tchl		↑ 300% SOD ↑ 388% POD ↑ 377% CAT		↑ 146%	In shoot: ↑ 531% Na^+^, ↓ 32% K^+^, ↓ 33% Mg^2+^, ↓ 27% Ca^2+^, Na^+^/K^+^ ratio ~8.6	
	10 ppm Boron + 150 mM NaCl + *B. pumilus*	↑ 3% SFW		↓ 5% Tchl		↑ 322% SOD ↑ 316% POD ↑ 254% CAT		↑ 85%	In shoot: ↑ 115% Na^+^, ↓ 11% K^+^, ↓ 5% Mg^2+^, ↓ 4% Ca^2+^, Na^+^/K^+^ ratio ~2.2	
*C. gleum* SUK. Wheat plantlets inoculated with *C. gleum.*	0 mM NaCl + Uninoculated									[99]
	100 mM NaCl + Uninoculated	↓ 41% SL ↓ 46% RL ↓ 16% TPFW	N/A	↓ 36% Tchl	N/A	N/A	↑ 80% FLA	↑ 31%	In shoot: ↑ 128% Na^+^, ↓ 30% K^+^, Na^+^/K^+^ ratio ~0.12	
	0 mM NaCl + SUK + FLI	↓ 13% SL ↓ 14% RL ↓ 0% TPFW		↑ 18% Tchl			↑ 96% FLA	↑ 48%	N/A	
	100 mM NaCl + SUK + FLI	↓ 9% SL ↓ 9% RL ↑ 19% TPFW		↑ 5% Tchl			↑ 147% FLA	↑ 63%	In shoot: ↑ 61% Na^+^, ↓ 19% K^+^, Na^+^/K^+^ ratio ~0.08	
	0 mM NaCl + SUK	↓ 16% SL ↓36% RL ↓13% TPFW		↓ 11% Tchl			↑ 57% FLA	↑ 25%	N/A	
	100 mM NaCl + SUK	↓ 19% SL, ↓ 9% RL, ↓ 6% TPFW		↓ 23% Tchl			↑ 84% FLA	↑ 47%	In shoot: ↑ 67% Na^+^, ↓ 19% K^+^, Na^+^/K^+^ ratio 0.08	
*B. aquimaris* DY-3. Three-day-old maize seedlings inoculated with DY-3	Control									[42]
	1% (*w*/*v*) NaCl + Uninoculated	↓ 34% TPDW	N/A	↓ 13% Tchl	↑ 39%	↑ 21% SOD ↑ 16% POD ↑18% CAT ↑ 23% APX	↑ 22% PHE	↑ 36%	N/A	
	0% NaCl + DY-3	↑ 12% TPDW		↑ 5% Tchl	↓ 8%	↑ 13% SOD ↑ 9% CAT ↑ 9% APX ↓ 12% POD	↑ 11% PHE	↑ 24%		
	1% (*w*/*v*) NaCl + DY-3	↓ 13% TPDW		↓ 9% Tchl	↑ 26%	↑ 53% SOD ↑ 42% CAT ↑ 65% APX ↓ 2% POD	↑ 67% PHE	↑ 77%		
*Klebsiella* IG3. Oat seedlings inoculated with IG3	Control									[44]
	100 mM NaCl + Uninoculated	↓ 22% SL ↓ 31% SFW ↓ 29% RFW ↓ 18% RL	↓ 6% IAA	↓ 22% Tchl	In shoot: ↑ 135% In root: ↑ 231%	↑ 353% SOD ↑ 540% POD	N/A	↑ 230%	N/A	
	0 mM NaCl + IG3	↑ 3% SL ↑ 3% SFW ↑ 1% RFW ↑ 13% RL	↑ 41% IAA	↑ 4% Tchl	In shoot: ↑ 3% In root: ↑ 18%	↑ 0% SOD ↑ 2% POD		↑ 42%		
	100 mM NaCl + IG3	↓ 10% SL, ↓ 18% SFW ↓ 16% RFW ↓ 2% RL	↑ 67% IAA	↓ 13% Tchl	In shoot: ↑ 27% In root: ↑ 45%	↑ 96% SOD ↑ 286% POD		↑ 155%		
*P. yonginensis* DCY84. Root seedlings of ginseng inoculated with *P. yonginensis* DCY84	Short period of stress (3 days of 300 mM NaCl exposure)									[50]
	Control								In shoot: Na^+^/K^+^ ratio ~13 In root: Na^+^/K^+^ ratio ~11	
	0 mM NaCl + DCY84	↑ 15% SFW ↑ 5% RFW	N/A	↑ 3% Chl a ↑ 2% Chl b ↑ 2% Car	Unchanged	↑ 62% APX ↑ 40% POD ↑ 114% CAT	N/A	↑ 253%	In shoot: Na^+^/K^+^ ratio ~15 In root: Na^+^/K^+^ ratio ~11	
	300 mM NaCl + Uninoculated	↓ 13% SFW ↓ 9% RFW		↓ 15% Chl a ↓ 13% Chl b ↓ 16% Car	↑ 29%	↑ 55% POD ↑ 0% APX ↓ 14% CAT		↑ 20%	In shoot: Na^+^/K^+^ ratio ~6.4 In root: Na^+^/K^+^ ratio ~7	
	300 mM NaCl + DCY84	↑ 12% SFW ↑ 0% RFW		↑ 3% Chl a ↓ 2% Chl b ↓ 9% Car	↑ 21%	↑ 54% APX ↑ 80% POD ↑ 114% CAT		↑ 233%	In shoot: Na^+^/K^+^ ratio ~8.2 In root: Na^+^/K^+^ ratio ~9.8	
	Long period of stress (12 days of 300 mM NaCl exposure)									
	Control								In shoot: Na^+^/K^+^ ratio ~7.2; In root: Na^+^/K^+^ ratio ~9.9	
	0 mM NaCl + DCY84	↑ 17% SFW ↑1% RFW	N/A	↑ 3% Chl a ↑ 2% Chl b ↓ 2% Car	Unchanged	↑ 45% APX ↑ 100% POD ↑ 143% CAT	N/A	↑ 300%	In shoot: Na^+^/K^+^ ratio ~7.8, In root: Na^+^/K^+^ ratio ~11	
	300 mM NaCl + Uninoculated	↓ 18% SFW ↓22% RFW		↓ 53% Chl a ↓ 66% Chl b ↓ 57% Car	↑ 36%	↓ 31% APX ↓ 33% POD ↓ 71% CAT		↑ 13%	In shoot: Na^+^/K^+^ ratio ~4.4, In root: Na^+^/K^+^ ratio ~3.8	
	300 mM NaCl + DCY84	↑ 12% SFW ↓ 3% RFW		↓ 3% Chl a ↓ 11% Chl b ↓ 7% Car	↑ 14%	↑ 90% APX ↑ 317% POD ↑ 343% CAT		↑ 287%	In shoot: Na^+^/K^+^ ratio ~6.5, In root: Na^+^/K^+^ ratio ~7.7	
*B. megaterium* A12 (BMA12). Ten-day-old tomato seedlings inoculated with BMA12	Control									[45]
	200 mM NaCl + Uninoculated	↓ 37% PH ↓ 50% RL ↓ 59% TPFW ↓ 54% TPDW ↓ 35% TLA	↓ 32% IAA ↓ 43% GA4 ↑ 100% C_2_H_4_ ↑ 82% ABA	↓ 32% Chl a ↓ 40% Chl b ↓ 41% TChl ↓ 49% Car	N/A	↑ 24% SOD ↑ 27% CAT ↓ 24% APX ↓ 24% POD ↓ 57% PPO	↑ 74% GSH ↑ 228% ASC	N/A	N/A	
	0 mM NaCl + BMA12	↑ 23% PH ↑ 37% RL ↑ 28% TPFW ↑ 40% TPDW ↑ 25% TLA	↑ 53% IAA, ↑ 170% GA4 ↓ 16% C_2_H_4_ ↓ 14% ABA	↑ 53% Chl a ↑ 14% Chl b ↑ 41% TChl ↑ 35% Car		↑ 86% SOD ↑ 54% CAT ↑ 34% APX ↑ 37% POD ↑ 55% PPO	↑ 17% GSH ↑ 5% ASC			
	2000 mM NaCl + BMA12	↓ 13% PH ↓ 28% RL ↓ 33% TPFW ↓ 35% TPDW ↓ 21% TLA	↑ 0% IAA ↑ 11%C_2_H_4_ ↑186% ABA, ↑ 86% GA4	↑ 5% Chl a ↓ 17% Chl b ↓ 4% TChl ↓ 24% Car		↑ 213% SOD ↑ 91% CAT ↑ 78% APX ↑ 18% POD ↓ 10% PPO	↑ 250% GSH ↑ 100% ASC			
*Pseudomonas* (wild-type UW4 and mutant strains). Seven-day-old tomato plants inoculated with UW4.	Control									[46]
	200 mM NaCl + Uninoculated	↓ 56% RL ↓ 37% SL ↓ 37% TPDW	N/A	↓ 42% Tchl	N/A					
	200 mM NaCl + WT UW4	↑ 16% RL ↑ 3% SL ↑ 25% TPDW		↑ 31% Tchl						
	200 mM NaCl + acdS-mutant	↓ 33% RL ↓ 9% SL ↓ 17% TPDW		↓ 25% Tchl						
	200 mM NaCl + treS- mutant	↓ 39% RL ↓ 31% SL ↓ 8% TPDW		↓ 13% Tchl						
	200 mM NaCl + acdS-/treS- double mutant	↓ 58% RL ↓ 37% SL ↓ 35% TPDW		↓ 56% Tchl						
	200 mM NaCl + OxtreS	↑ 45% RL ↑ 3% SL ↑ 54% TPDW		↑ 61% Tchl						
*S. maltophilia* BJ01. Seven-day-old peanut seedlings inoculated with BJ01	Control									[48]
	0 mM NaCl + BJ01	↑ 4% SL, ↑ 11% TPFW, ↓ 15% RL	↑ 19% Aux	↑ 11% Chla, ↑ 0% Chl b, ↑ 17% Tchl	↓ 26%	N/A		↓ 32%	N/A	
	100 mM NaCl + Uninoculated	↑ 9% RL, ↓ 45% TPFW, ↓ 39% SL	↑ 16% Aux	↓ 56% Chl a, ↓ 42% Chl b, ↓ 50% Tchl	↑ 47%			↑ 1355%		
	100 mM NaCl + BJ01	↓ 26% SL, ↓ 3% RL, ↓ 26% TPFW	↑ 29% Aux	↓ 11% Chl a ↓ 34% Chl b ↓ 23% Tchl	↑ 16%			↑ 1173%		

Note: All calculations in the Table 1 represent the comparisons between the treated plants and the control plants (non-stress conditions and un-inoculation). The up arrowhead (↑) indicates an increase in a tested parameter as compared to the control. The down arrowhead (↓) displays a reduction in a tested parameter relative to the control. Abbreviation in the Table 1: ABA, Abscisic acid; *A. calcoaceticus*, *Acinetobacter calcoaceticus*; *A. aneurinilyticus*, *Aneurinibacillus aneurinilyticus*; AEs, antioxidant enzymes; APX, Ascorbate peroxidase; ASC, Ascorbate; Aux, Auxin; *B. aquimaris*, *Bacillus aquimaris*; *B. brevis*, *Bacillus brevis*; *B. cepacia*, *Burkholdera cepacia*; *B. megaterium*, *Bacillus megaterium*; *B. pumilus*, *Bacillus pumilus*; C_2_H_4_, ethylene; Car, Carotenoids; CAT, Catalase; *C. gleum*, *Chryseobacterium gleum*; Chl, Chlorophyll; *E. aerogenes*, *Enterobacter aerogenes*; FLA, Flavonoids; FLI, Feather lysate inoculum; GA4, Gibberellins 4; GSH, Glutathione; GP, Growth parameter; IAA, Indole-3-acetic acid; JA, Jasmonates; MDA, Malondialdehyde; N/A, Not available; NEAs, Non-enzymatic antioxidants; *P. fluorescence*, *Pseudomonas fluorescence*; PH, Plant height; PHE, Phenols; PhoPs, Photosynthetic pigments; POD, Peroxidase; PPs, Polyphenols; Pro, Proline; *P. putida*, *Pseudomonas putida*; PPO, Polyphenol oxidase; Pro, Proline; *P. yonginensis*, *Paenibacillus yonginensis*; RDW, Root dry weight; RFW, Root fresh weight; RL, Root length; SA, Salicylic acid; SDW, Shoot dry weight; SFW, Shoot fresh weight; SL, Shoot length; *S. maltophilia*, *Stenotrophomonas maltophilia*; SOD, Superoxide dismutase; Tchl, Total chlorophyll; TLA, Total leaves area per plant; TPDW, Total plant dry weight; TPFW, Total plant fresh weight; TPP, Total polyphenol; *X. autotrophicus*, *Xanthobacter autotrophicus*; Y, Yield.

**Table 2 ijms-22-03154-t002:** The use of PGPR in seed biopriming technique for improving salinity stress tolerance in plants.

PGPR	Treatments	GP	PhoPs	AEs	MDA	Ion Content	Pro	Ethylene	Sources
*S. maltophilia* SBP-9. Bacterized wheat seeds with *S. maltophilia* SBP-9 for 1 h.	Control								[102]
	0 mM NaCl + SBP-9	↑ 15% SL, ↑ 10% RL, ↑ 12% SFW, ↑ 17% SDW, ↑ 33% RFW, ↑ 9% RDW	↑ 8% Tchl	↑ 33% SOD ↑ 20% CAT ↑ 33% POD	↓ 27%	In shoot: ↓ 4% Na^+^, ↑ 12% K^+^, Na^+^/K^+^ ratio ~0.54	↓ 10%	N/A	
	150 mM NaCl + Unprimed	↓ 11% SL, ↓ 5% RL, ↓ 8% SFW, ↓ 24% SDW, ↓ 11% RFW, ↓ 23% RDW	↓ 15% Tchl	↑ 58% SOD ↑ 7% CAT ↑ 67% POD	↑ 17%	In shoot: ↑ 48% Na^+^, ↓ 17% K^+^, Na^+^/K^+^ ratio ~1.12	↑ 74%		
	150 mM NaCl + SBP-9	↑ 4% SL, ↑ 15% RL, ↑ 4% SFW, ↑ 7% SDW, ↑ 22% RFW, ↓ 5% RDW	↑ 5% Tchl	↑ 133% SOD ↑ 93% CAT ↑ 133% POD	↓ 13%	In shoot: ↑ 20% Na^+^, ↑ 3% K^+^ Na^+^/K^+^ ratio ~0.73	↑ 23%		
	200 mM NaCl + Unprimed	↓ 37% SL, ↓ 20% RL, ↓ 32% SFW, ↓ 38% SDW, ↓ 33% RFW, ↓ 64% RDW	↓ 59% Tchl	↑ 217% SOD ↑ 100% CAT ↑ 192% POD	↑ 93%	In shoot: ↑ 107% Na^+^, ↓ 25% K^+^, Na^+^/K^+^ ratio ~1.73	↑ 165%		
	200 mM NaCl + SBP-9	↓ 11% SL, ↓ 5% RL, ↓ 16% SFW, ↓ 17% SDW, ↓ 6% RFW, ↓ 41% RDW	↓ 39% Tchl	↑ 350% SOD ↑ 180% CAT ↑ 283% POD	↑ 50%	In shoot: ↑ 54% Na^+^, ↓ 3% K^+^ Na^+^/K^+^ ratio ~1	↑ 110%		
*Enterobacter* P23. Seeds of *Oryza sativa* cv. Ratna treated with bacterial suspension.	Control								[49]
	150 mM NaCl + Unprimed	↓ 51% GP, ↓ 97% SVI, ↓ 45% SFW, ↓ 58% SDW, ↓ 33% SL, ↓ 39% RFW, ↓ 63% RDW, ↓ 44% RL	↓ 54% Chl a ↓ 80% Chl b ↓ 56% Tchl	↑ 120% SOD ↑ 112% CAT ↑ 174% POD ↑ 700% PPO	↑ 300%	N/A	↑ 175%	N/A	
	150 mM NaCl + P23	↓ 22% GP, ↓ 58% SVI, ↓ 16% SFW, ↓ 11% SL, ↓ 23% SDW, ↓ 15% RFW, ↓ 30% RDW, ↓ 11% RL	↓ 13% Chl a ↓ 10% Chl b ↓ 8% Tchl	↑ 32% SOD ↑ 46% CAT ↑ 70% POD ↑ 300% PPO	↑ 195%		↑ 75%		
*B. pumilus* FAB10. Wheat seeds cv. 343 treated with FAB10.	Control								[51]
	75 mM NaCl + Unprimed	↓ 17% SL, ↓ 35% RL, ↓ 49% SDW, ↓ 53% RDW, ↓ 35% SpDW, ↓ 21% GY, ↓ 17% GPr	N/A	↑ 20% SOD ↑ 40% CAT ↑ 50% GR	↑ 189%	N/A	↑ 105%		
	125 mM NaCl + Unprimed	↓ 24% SL, ↓ 49% RL, ↓ 42% SDW, ↓ 67% RDW, ↓ 48% SpDW, ↓ 31% GY, ↓ 22% GPr		↑ 23% SOD ↑ 80% CAT ↑ 75% GR	↑ 189%		↑ 146%		
	250 mM NaCl + Unprimed	↓ 35% SL, ↓ 52% RL, ↓ 40% SDW, ↓ 76% RDW, ↓ 61% SpDW, ↓ 41% GY, ↓ 29% GPr		↑ 25% SOD ↑ 80% CAT ↑ 75% GR	↑ 260%		↑ 171%		
	75 mM NaCl + FAB10	↓ 13% SL, ↓ 11% RL, ↓ 13% SDW, ↓ 20% RDW, ↓ 4% SpDW, ↓ 3% GY, ↓ 11% GPr		↑ 5% SOD ↓ 20% CAT ↓ 25% GR	↑ 103%		↑ 77%		
	125 mM NaCl + FAB10	↓ 17% SL, ↓ 22% RL, ↓ 18% SDW, ↓ 43% RDW, ↓ 9% SpDW, ↓ 13% GY, ↓ 16% GPr		↑ 10% SOD ↑ 20% CAT ↑ 0% GR	↑ 180%		↑ 123%		
	250 mM NaCl + FAB10	↓ 25% SL, ↓ 24% RL, ↓ 18% SDW, ↓ 51% RDW, ↓ 61% SpDW, ↓ 25% GY, ↓ 27% GPr		↑ 12% SOD ↑ 20% CAT ↑ 0% GR	↑ 237%		↑ 139%		
Consortium (*R. leguminosarum* + *A. chroococcum*) and/or Mel. *Vicia faba* seeds were treated with the consortium as peat-based inoculant and/or Mel solution	**Saline soil** Control								[124]
	25 µM Mel + Unprimed	↑ 6% SL, ↑ 37% NL, ↑ 18% SFW, ↑ 24% SDW, ↑ 23% Y	↑ 6% Chl a ↑ 11% Chl b ↑ 9% Car	N/A	N/A	N/A	↑ 17%	N/A	
	50 µM Mel + Unprimed	↑ 24% SL, ↑ 56% NL, ↑ 41% SFW, ↑ 36% SDW, ↑ 41% Y	↑ 30% Chl a ↑ 26% Chl b ↑ 18% Car				↑ 30%		
	100 µM Mel + Unprimed	↑ 41% SL, ↑ 93% NL, ↑ 72% SFW, ↑ 78% SDW, ↑ 58% Y	↑ 44% Chl a ↑ 80% Chl b ↑ 35% Car				↑ 39%		
	0 µM Mel + Primed	↑ 18% SL, ↑ 43% NL, ↑ 25% SFW, ↑ 36% SDW, ↑ 48% Y	↑ 31% Chl a ↑ 35% Chl b ↑ 28% Car				↑ 44%		
	25 µM Mel + Primed	↑ 30% SL, ↑ 79% NL, ↑ 49% SFW, ↑ 56% SDW, ↑ 71% Y	↑ 42% Chl a ↑ 59% Chl b ↑ 43% Car				↑ 57%		
	50 µM Mel + Primed	↑ 70% SL, ↑ 97% NL, ↑ 68% SFW, ↑ 68% SDW, ↑ 82% Y	↑ 56% Chl a ↑ 107% Chl b ↑ 66% Car				↑ 89%		
	100 µM Mel + Primed	↑ 74% SL, ↑ 118% NL, ↑ 98% SFW, ↑ 104% SDW, ↑ 96% Y	↑ 71% Chl a ↑ 118% Chl b ↑ 71% Car				↑ 110%		
	**Non-saline soil** Control								
	25 µM Mel + Unprimed	↑ 6% SL, ↑ 14% NL, ↑ 10% SFW, ↑ 21% SDW, ↑ 18% Y	↑ 9% Chl a ↑ 10% Chl b ↑ 4% Car				↑ 5%		
	50 µM Mel + Unprimed	↑ 12% SL, ↑ 35% NL, ↑ 23% SFW, ↑ 31% SDW, ↑ 29% Y	↑ 11% Chl a ↑ 24% Chl b ↑ 21% Car				↑ 11%		
	100 µM Mel + Unprimed	↑ 16% SL, ↑ 49% NL, ↑ 38% SFW, ↑ 69% SDW, ↑ 32% Y	↑ 16% Chl a ↑ 34% Chl b ↑ 24% Car				↑ 45%		
	0 µM Mel + Primed	↑ 20% SL, ↑ 28% NL, ↑ 14% SFW, ↑ 27% SDW, ↑ 17% Y	↑ 17% Chl a ↑ 24% Chl b ↑ 21% Car				↑ 36%		
	25 µM Mel + Primed	↑ 29% SL, ↑ 49% NL, ↑ 24% SFW, ↑ 46% SDW, ↑ 25% Y	↑ 23% Chl a ↑ 30% Chl b ↑ 30% Car				↑ 94%		
	50 µM Mel + Primed	↑ 45% SL, ↑ 55% NL, ↑ 33% SFW, ↑ 57% SDW, ↑ 38% Y	↑ 26% Chl a ↑ 42% Chl b ↑ 33% Car				↑ 110%		
	100 µM Mel + Primed	↑ 49% SL, ↑ 76% NL, ↑ 52% SFW, ↑ 87% SDW, ↑ 45% Y	↑ 30% Chl a ↑ 29% Chl b ↑ 40% Car				↑ 139%		
*A. aneurinilyticus* ACC02, *Paenibacillus* ACC06 and Consortium (ACC02+ ACC06). French bean seeds inoculated with ACC02, ACC06 and consortium (ACC02 + ACC06)	Control								[88]
	0 mM NaCl + ACC02	↑ 10% SL, ↑ 50% RL, ↑ 158% SFW, ↑ 10% SDW, ↑ 50% RFW, ↑ 21% RDW	↑ 36% Tchl	N/A	N/A	N/A	N/A	↑ 9%	
	0 mM NaCl + ACC06	↑ 30% SL, ↑ 30% RL, ↑ 216% SFW, ↑ 10% SDW, ↑ 60% RFW, ↑ 14% RDW	↑ 29% Tchl					↓ 9%	
	0 mM NaCl + Consortium	↑ 50% SL, ↑ 70% RL, ↑ 233% SFW, ↑ 80% SDW, ↑ 90% RFW, ↑ 85% RDW	↑ 57% Tchl					↑ 27%	
	25 mM NaCl + Unprimed								
	25 mM NaCl + ACC02	↑ 33% SL, ↑ 79% RL, ↑ 120% SFW, ↑ 300% SDW, ↑ 46% RFW, ↑ 182% RDW	↑ 28% Tchl	N/A	N/A	N/A	N/A	↓ 38%	
	25 mM NaCl + ACC06	↑ 47% SL, ↑ 58% RL, ↑ 120% SFW, ↑ 350% SDW, ↑ 36% RFW, ↑ 142% RDW	↑ 35% Tchl					↓ 42%	
	25 mM NaCl + Consortium	↑ 60% SL, ↑ 110% RL, ↑ 255% SFW, ↑ 425% SDW, ↑ 81% RFW, ↑ 220% RDW	↑ 57% Tchl					↓ 61%	
*P. fluorescence,* *B. pumilus, E. aurantiacum* and consortium (*P. fluorescence* + *B. pumilus* + *E.aurantiacum*) Wheat seeds soaked in bacterial inoculant containing single PGPR strains or the consortium of three bacterial cultures for 2 h. Saline soil EC_e_ 13.41	Unprimed seeds								[122]
	Seeds primed with *P. fluorescence*	**Galaxy-13:** ↑ 5% SL, ↑ 7% RL, ↑ 3% SFW, ↑ 2% SDW, ↑ 33% 100 GW, ↓ 13% RFW, ↓ 29% RDW **Aas-11:** ↑ 11% SL, ↑ 24% RL, ↑ 48% SFW, ↑ 144% RFW, ↑ 57% SDW, ↑ 75% RDW, ↑ 23% 100 GW	N/A	**Galaxy-13:** ↓ 30% SOD ↓ 0% POD ↑ 27% CAT **Aas-11:** ↓ 57% SOD ↓ 14% POD ↑ 25% CAT	N/A	**Galaxy-13:** **In root:** ↑ 50% Na^+^, ↑ 40% K^+^, Na^+^/K^+^ ratio ~0.19 **In shoot:** ↑ 28% Na^+^, ↑ 23% K^+^, Na^+^/K^+^ ratio ~3.9 **Aas-11:** **In root:** ↓ 13% Na^+^, ↑ 99% K^+^, Na^+^/K^+^ ratio ~0.16. **In shoot:** ↑ 92% Na^+^, ↑ 16% K^+^, Na^+^/K^+^ ratio ~3.4	**Galaxy-13:** ↓ 20% **Aas-11:** ↑ 33%	N/A	
	Seeds primed with *B. pumilus*	**Galaxy-13:** ↓ 7% SL, ↓ 18% SFW, ↓ 26% SDW, ↓ 8% RFW, ↓ 57% RDW, ↑ 67% RL, 31% 100 GW **Aas-11:** ↑ 13% SL, ↑ 21% RL, ↑ 61% SFW, ↑ 678% RFW, ↑ 66% SDW, ↑ 838% RDW, ↑ 53% 100 GW		**Galaxy-13:** ↓ 35% SOD ↓ 5% POD ↑ 4% CAT **Aas-11:** ↓ 65% SOD ↓ 38% POD ↑ 35% CAT		**Galaxy-13:** **In root:** ↑ 0% Na^+^, ↑ 34% K^+^, Na^+^/K^+^ ratio ~0.13 **In shoot:** ↓ 8% Na^+^, ↓ 19% K^+^, Na^+^/K^+^ ratio ~4.3 **Aas-11:** **In root:** ↓ 13% Na^+^, ↑ 195% K^+^, Na^+^/K^+^ ratio ~0.11. **In shoot:** ↑ 59% Na^+^, ↓ 12% K^+^, Na^+^/K^+^ ratio ~3.7	**Galaxy-13:** ↑ 287% **Aas-11:** ↑ 150%		
	Seeds primed with *E.aurantiacum*	**Galaxy-13:** ↑ 6% SL, ↑ 47% RL, ↑ 3% SFW, ↑ 49% 100 GW, ↓ 17% RFW, ↓ 2% SDW, ↓ 28% RDW **Aas-11:** ↑ 10% SL, ↑ 7% RL, ↑ 52% SFW, ↑ 511% RFW, ↑ 71% SDW, ↑ 713% RDW, ↑ 47% 100 GW		**Galaxy-13:** ↑ 2% SOD ↑ 48% CAT ↓ 43% POD **Aas-11:** ↓ 65% SOD ↓ 57% POD ↓ 5% CAT		**Galaxy-13:** **In root:** ↑ 33% Na^+^, ↑ 34% K^+^, Na^+^/K^+^ ratio ~0.18 **In shoot:** ↑ 27% Na^+^, ↑ 0% K^+^, Na^+^/K^+^ ratio ~4.77. **Aas-11:** In root: ↓ 13% Na^+^, ↑ 286% K^+^, Na^+^/K^+^ ratio ~0.08 **In shoot:** ↑ 40% Na^+^, ↑ 22% K^+^, Na^+^/K^+^ ratio ~2.36	Galaxy-13: ↑ 227% Aas-11: ↑ 110%		
	Seeds primed with a consortium	**Galaxy-13:** ↓ 1% SL, ↑ 73% RL, ↑ 6% SFW, ↑ 30% RFW, ↑ 7% SDW, ↑ 43% RDW, ↑ 53% 100 GW **Aas-11:** ↑ 13% SL, ↑ 3% RL, ↑ 65% SFW, ↑ 556% RFW, ↑ 77% SDW, ↑ 725% RDW, ↑ 48% 100 GW		**Galaxy-13:** ↑ 37% SOD ↓ 32% POD ↓ 6% CAT **Aas-11:** ↓ 57% SOD ↑ 24% POD ↑ 28% CAT		**Galaxy-13:** **In root:** ↑ 0% Na^+^, ↑ 114% K^+^, Na^+^/K^+^ ratio ~0.08 **In shoot:** ↑ 17% Na^+^, ↑ 15% K^+^, Na^+^/K^+^ ratio ~3.8 **Aas-11:** **In root:** ↑ 0% Na^+^, ↑ 173% K^+^, Na^+^/K^+^ ratio ~0.13 **In shoot:** ↑ 68% Na^+^, ↑ 30% K^+^, Na^+^/K^+^ ratio ~2.67	**Galaxy-13:** ↑ 327% **Aas-11:** ↑ 17%		
*Sphingobacterium* BHU-AV3. Bacterized tomato seeds with BHU-AV3 for 24h	Control								[47]
	200 mM NaCl + Unprimed	↓ 52% SL, ↓ 49% RL, ↓ 54% TPDW	↓ 44% Tchl	In shoot: ↑ 90% SOD ↑ 260% POD ↑ 100% PPO In root: ↑ 83% SOD ↑ 100% POD ↑ 53% PPO	N/A	In shoot: ↑ 258% Na^+^, ↓ 63% K^+^, Na^+^/K^+^ ratio ~3.6 In root: ↑ 190% Na^+^, ↓ 53% K^+^, Na^+^/K^+^ ratio ~3.5	In shoot: ↑ 153% In root: ↑ 56%	N/A	
	0 mM NaCl + BHU-AV3	↓ 11.3% SL, ↑ 16% RL, ↑ 11% TPDW	↑ 5% Tchl	In shoot: ↓ 20% SOD ↓ 0% POD ↓ 25% PPO In root: ↑ 16% SOD ↑ 6% POD ↓ 12% PPO		In shoot: ↑ 9% Na^+^, ↑ 9% K^+^, Na^+^/K^+^ ratio ~0.3 In root: ↓ 5% Na^+^, ↑ 8% K^+^, Na^+^/K^+^ ratio ~0.5	In shoot: ↓ 7% In root: ↓ 5%		
	200 mM NaCl + BHU-AV3	↓ 30% SL, ↓ 22% RL, ↓ 29% TPDW	↓ 14% Tchl	In shoot: ↑ 10% SOD ↑ 1000% POD ↑ 50% PPO In root: ↑ 117% SOD ↑ 200% POD ↑ 71% PPO		In shoot: ↑ 130% Na^+^, ↓ 20% K^+^, Na^+^/K^+^ ratio ~1 In root: ↑ 115% Na^+^, ↓ 24% K^+^, Na^+^/K^+^ ratio ~1.6	In shoot: ↑ 84% In root: ↑ 111%		
*K. sacchari* MSK1. Mung bean seeds primed with MSK1	Control								[36]
	50 mM NaCl + Unprimed	↓ 8% SL, ↓ 8% RL, ↓ 5% SDW, ↓ 5% RDW, ↓ 15% SY, ↓ 3% GP	↓ 15% Tchl ↓ 4% Car	↑ 5% GR, ↑ 33% CAT ↑ 13% SOD ↑ 23% APX	↑ 32%	In shoor: ↑ 100% Na^+^, ↑ 44% K^+^, Na^+^/K^+^ ratio ~0.46, ↓ 4% N, ↓ 19% P	↑ 63%	N/A	
	100 mM NaCl + Unprimed	↓ 16% SL, ↓ 20% RL, ↓ 10% SDW, ↓ 15% RDW, ↓ 21% SY, ↓ 6% GP	↓ 35% Tchl ↓ 9% Car	↑ 15% GR ↑ 58% CAT ↑ 39% SOD ↑ 45% APX	↑ 47%	In shoot: ↑ 200% Na^+^, ↑ 100% K^+^, Na^+^/K^+^ ratio ~0.5, ↓ 4% N, ↓ 28% P	↑ 88%		
	200 mM NaCl + Unprimed	↓ 24% SL, ↓ 28% RL, ↓ 21% SDW, ↓ 35% RDW, ↓ 26% SY, ↓ 8% GP	↓ 42% Tchl ↓ 19% Car	↑ 35% GR ↑ 108% CAT ↑ 52% SOD ↑ 73% APX	↑ 84%	In shoot: ↑ 450% Na^+^, ↑ 222% K^+^, Na^+^/K^+^ ratio ~0.57, ↓ 12% N, ↓ 44% P	↑ 213%		
	400 mM NaCl + Unprimed	↓ 41% SL, ↓ 52% RL, ↓ 34% SDW, ↓ 55% RDW, ↓ 34% SY, ↓ 26% GP	↓ 62% Tchl ↓ 33% Car	↑ 64% GR ↑ 208% CAT ↑ 96% SOD ↑ 102% APX	↑ 153%	In shoot: ↑ 800% Na^+^, ↑ 378% K^+^, Na^+^/K^+^ ratio ~0.63, ↓ 21% N, ↓ 59% P	↑ 350%		
	0 mM NaCl + MSK1	↑ 5% SL, ↑ 12% RL, ↑ 7% SDW, ↑ 15% RDW, ↑ 9% SY, ↑ 7% GP	↑ 29% Tchl ↑ 7% Car	↓ 9% GR ↓ 33% CAT ↓ 22% SOD ↓ 9% APX	↓ 37%	In shoot: ↓ 67% Na^+^, ↓ 22% K^+^, Na^+^/K^+^ ratio ~0.14, ↑ 9% N, ↑ 15% P	↓ 25%		
	50 mM NaCl + MSK1	↓ 3% SL, ↑ 4% RL, ↓ 2% SDW, ↑ 3% RDW, ↑ 10% SY, ↑ 2% GP	↓ 3% Tchl ↓ 0% Car	↑ 2% GR ↑ 8% CAT ↑ 9% SOD ↑ 9% APX	↑ 11%	In shoot: ↑ 33% Na^+^, ↑ 22% K^+^, Na^+^/K^+^ ratio ~0.36, ↓ 1% N, ↓ 41% P	↑ 30%		
	100 mM NaCl + MSK1	↓ 8% SL, ↓ 12% RL, ↓ 7% SDW, ↓ 10% RDW, ↓ 19% SY, ↓ 5% GP	↓ 31% Tchl ↓ 8% Car	↑ 11% GR ↑ 50% CAT ↑ 22% SOD ↑ 32% APX	↑ 37%	In shoot: ↑ 183% Na^+^, ↑ 89% K^+^, Na^+^/K^+^ ratio ~0.5, ↓ 4% N, ↓ 22% P	↑ 75%		
	200 mM NaCl + MSK1	↓ 22% SL, ↓ 16% RL, ↓ 17% SDW, ↓ 25% RDW, ↓ 24% SY, ↓ 7% GP	↓ 35% Tchl ↓ 13% Car	↑ 33% GR ↑ 100% CAT ↑ 48% SOD ↑ 64% APX	↑ 79%	In shoot: ↑ 433% Na^+^, ↑ 211% K^+^, Na^+^/K^+^ ratio ~0.57, ↓ 9% N, ↓ 37% P	↑ 200%		
	400 mM NaCl + MSK1	↓ 35% SL, ↓ 24% RL, ↓ 32% SDW, ↓ 48% RDW, ↓ 32% SY, ↓ 25% GP	↓ 54% Tchl ↓ 27% Car	↑ 60% GR ↑ 192% CAT ↑ 91% SOD ↑ 91% APX	↑ 137%	In shoot: ↑ 783% Na^+^, ↑ 367% K^+^, Na^+^/K^+^ ratio ~0.63, ↓ 19% N, ↓ 57% P	↑ 325%		

Note: All calculations in the Table 2 represent the comparisons between the treated plants and the control plants (non-stress conditions and un-priming) except that the control plants in the study of Nawaz et al. (2020) [122] were cultivated in the saline soil ECe ~13.41. The up arrowhead (↑) indicates an increase in a tested parameter as compared to the control. The down arrowhead (↓) displays a reduction in a tested parameter relative to the control. Abbreviation in the Table 2: *A.*
*calcoaceticus*, *Acinetobacter calcoaceticus*; *A. aneurinilyticus*, *Aneurinibacillus aneurinilyticus*; *A. chroococcum*, *Azotobacter chroococcum*; AEs, Antioxidant enzymes; APX, Ascorbate peroxidase; *B. pumilus*, *Bacillus pumilus*; Car, Carotenoids; CAT, Catalase; Chl, Chlorophyll; *E. aurantiacum*, *Exiguobacterium aurantiacum*; GP, Germination percentage; GPr, Grain protein; GR, Glutathione reductase; GW, Grain weight; GY, Grain yield; *K. sacchari*, *Kosakonia sacchari*; MDA, Malondialdehyde; Mel, Melatonin; N/A, Not available; NL, Number of leaves per plant; *P.*
*fluorescence*, *Pseudomonas fluorescence*; PhoPs, Photosynthetic pigments; POD, Peroxidase; *P.*
*putida*, *Pseudomonas putida*; PPO, Polyphenol oxidase; RDW, Root dry weight; RFW, Root fresh weight; *R. leguminosarum*, *Rhizobium leguminosarum*; RL, Root length; SDW, Shoot dry weight; SFW, Shoot fresh weight; SL, Shoot length; *S. maltophilia*, *Stenotrophomonas maltophilia*; SOD, Superoxide dismutase; SpDW, Spike dry weight; SVI, Seedling vigor index; SY, Seed yield; Tchl, Total chlorophyll; TPDW, Total plant dry weight; Y, Yield.

## Data Availability

Not applicable.

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
