# Peer review of "Roles of Plant Growth-Promoting Rhizobacteria (PGPR) in Stimulating Salinity Stress Defense in Plants: A Review"

_ijms, 2021, doi:10.3390/ijms22063154_

Round 1

Reviewer 1 Report

In general the review revisits role of PGPR in salinity stress and highlights the effect on plant growth. Although the review is well written and detailed, I find it repetitive and just an update with few new references included. The review presents the already known approaches and mechanisms.

I would suggest authors to focus on the molecular aspects which are trigerred by PGPR and elucidate it further with their own viewpoints.

Reviewer 2 Report

The document "Roles of plant growth-promoting rhizobacteria (PGPR) in stimulating salinity stress defense in plants: A review"  is particularly interesting.

Describes a topical issue for agriculture. Have a very nice figure 1.

There are several review articles on PGRR in the international literature. There are a few things I suggest to do in order to improve the document.

  1. Write an independent paragraph on the relationship between the root system and PGPR
  2. The tables are too big and not easy for the reader. Fot Example Table 2 is 10 pages for a description of 9 reffereences . Table 1 is 8 pages. This is the weakest point of the document. It must be improved anyway.
  3. Species names must be written in italics. 

Reviewer 3 Report

This is an extensive survey on the effects of plant growth-promoting rhizobacteria (PGPR) on plant morphological and physiological aspects. However, the text is descriptive and often a critical holistic view is missing.

To my opinion, the largest controversy comes from the fact that PGPR include several excessively different bacteria, which obviously induce different effects on the plants. Therefore, by comparing the same bacteria genera or some related kind of grouping based on their properties may be a more efficient strategy of comparing different studies. This analysis is further complicated by the fact that these are added in the soil, which gives a huge variability regarding their proliferation. These aspects ought to be included in the manuscript.

What is the difference between promotive effect of PGPR on plants and germination? Can these effects being grouped? Can you highlight the similarities and the discrepancies?

There are repetitions throughout the manuscript that should be avoided. For instance, in one section you present the importance of ROS extensively, in another section the effect of PGPR under the absence of stress is discussed, and in a third section the effect of PGPR under salinity is provided. This brings repetitions.

There are controversial findings in the literature. What is the opinion of the authors? What should be taken into consideration for future experiments? Please present highlights for future work.

Line 34: “is caused” and no need to use the “by” three times (the first one would be sufficient

Lines 39, 110, 113: give a range of NaCl concentration in parenthesis when referring to high or moderate salinity levels

Line 48: give some examples of agricultural practices to deal with salinity [e.g. silicon (Hassanvand et al., 2019 Industrial Crops and Products 134, 19–25), melatonin (Park et al., 2021 Front. Plant Sci. doi: 10.3389/fpls.2021.593717)]

Lines 55-57: are these agricultural practices (e.g., pesticides) relevant to salt stress discussed here?

Sections starting at Lines 69, 119: a conclusion at the end of these sessions is missing. What the authors conclude based on the reported findings

Lines 82, 86, 87, 89, 94, 102, 109 and throughout the manuscript: species names and in vitro ought to be in italics

Lines 97-99: the effect of salt stress on photosynthetic pigments, ROS accumulation and several traits reported here is presented in detail in further sections. Thus, you need to erase them here to avoid repetition.

Line 116: the effect of salinity on essential oil yield of medicinal plants has also been studied (Hassanvand et al., 2019 Industrial Crops and Products 134, 19–25)

Session starting at line 142: what is the effect of salinity in these processes?

Conclusions’ session: no new ideas ought to be presented here. This session must be a short summary of what has been discussed. What is given here may be provided in an earlier section called Perspective for further research

Round 2

Reviewer 1 Report

Authors have taken into account most of the comments and have improved the manuscript accordingly. Specifically the addition of multi-omics perspective and also future directions is detailed and informative.

Reviewer 2 Report

the manuscript improved

I suggesti to accept it,

Reviewer 3 Report

My comments were respectfully addressed

Line 18: erase “in plants”

Line 28: do you use this abbreviation in the abstract? If not, erase it

Line 61: is this abbreviation (AE) earlier introduced?

Line 76: is this abbreviation earlier introduced?

Lines 161-162: since two studies were focused on tomato plants, then give the range of the effect (42-44%) and put the references together (46, 47)
